# Langevin Soft Actor-Critic: Efficient Exploration through Uncertainty-Driven Critic Learning

**Haque Ishfaq**[*]**, Guangyuan Wang**[*]**, Sami Nur Islam, Doina Precup**
Mila, McGill University
{haque.ishfaq, guangyuan.wang}@mail.mcgill.ca

## Abstract

Existing actor-critic algorithms, which are popular for continuous control reinforcement learning (RL) tasks, suffer from poor sample efficiency due to lack of principled exploration mechanism within them. Motivated by the success of Thompson sampling for efficient exploration in RL, we propose a novel model-free RL algorithm, *Langevin Soft Actor Critic* (LSAC), which prioritizes enhancing critic learning through uncertainty estimation over policy optimization. LSAC employs three key innovations: approximate Thompson sampling through distributional Langevin Monte Carlo (LMC) based $Q$ updates, parallel tempering for exploring multiple modes of the posterior of the $Q$ function, and diffusion synthesized state-action samples regularized with $Q$ action gradients. Our extensive experiments demonstrate that LSAC outperforms or matches the performance of mainstream model-free RL algorithms for continuous control tasks. Notably, LSAC marks the first successful application of an LMC based Thompson sampling in continuous control tasks with continuous action spaces.

## 1 Introduction

We introduce a practical and efficient model-free online RL algorithm termed *Langevin Soft Actor-Critic* (LSAC), which incorporates distributional Langevin Monte Carlo (LMC) (Welling & Teh, 2011) critic updates with parallel tempering and action refinement on diffusion synthesized trajectories. Our approach employs a distributional $Q$ objective and allows diverse sampling from multimodal $Q$ posteriors through the use of parallel tempering (Chandra et al., 2019), making LSAC especially well-suited for continuous control tasks in environments like MuJoCo control tasks (Brockman et al., 2016) and DeepMind Control Suite (DMC) (Tassa et al., 2018).

Although Langevin-style update is powerful for learning posteriors by performing noisy gradient descent updates to approximately sample from the exact posterior distribution of the $Q$ function, when naively applied to continuous control settings, it meets with the following three challenges:

(C1) **Multidimensional continuous action spaces.** In continuous control settings, actions are typically continuous and multidimensional tensors. This makes it computationally intractable to naively select the exact greedy actions based on $Q$ posterior approximations, often leading to sub-optimal performance and inefficient exploration.

(C2) **Value approximation errors.** While LMC update helps in better exploration (Ishfaq et al., 2024a;b), it may also lead to instability issues when coupled with deep neural networks (Dauphin et al., 2014) due to overestimation bias of $Q$-function. Moreover, naive LMC might lead to similar actions being overly explored due to high correlation among samples from the LMC Markov chain at nearby steps which in turn can lead to value approximation error (Holden, 2019; Vishnoi, 2021).

(C3) **Low sample efficiency.** Naive LMC updates, akin to many actor-critic frameworks, use a Update-To-Data (UTD) ratio of 1, which is the number of network updates to actual environment interactions. This limited UTD ratio often leads to underfitting the complex state-action

---

[*]Equal contribution

representations in continuous control (Chen et al., 2021; Dorka et al., 2023). Relying solely on a single critic update per iteration from on-policy experience is insufficient, as it fails to leverage the diversity of the data space, thereby hindering critic learning.

Recently there have been several works (Dwaracherla & Van Roy, 2020; Ishfaq et al., 2024a;b) that provide provably efficient RL algorithms that rely on LMC-style updates. However, these algorithms are scalable only to pixel-based Deep RL environments with discrete action spaces. One challenge arises from the multidimensional continuous action spaces in continuous control environments, as detailed in (C1). LSAC addresses this challenge by eliminating the need to compute the maximum of $Q$ values over the entire action space to select greedy actions (Ishfaq et al., 2024a). Instead, it employs a distributional critic learning framework using LMC along with a Maximum Entropy (Max-Ent) policy objective (Eysenbach & Levine, 2022). Learning distributional critic further mitigates $Q$-value overestimation issue as detailed in (C2). To address (C3), LSAC further incorporates $Q$ action gradient refinement (Yang et al., 2023) in diffusion synthesized state-action samples during the critic update. This introduces diverse and potentially high-valued synthetic state-action pairs into the collected trajectories, thereby enhancing critic learning. The actor in turn benefits from more accurate value estimations and thus improved policy learning. We further use parallel tempering (Chandra et al., 2019) to allow sampling from multi-modal $Q$ function posterior.

Furthermore, traditional continuous control benchmarks such as DSAC-T (Duan et al., 2023), REDQ (Chen et al., 2021), SAC (Haarnoja et al., 2018a), and TD3 (Fujimoto et al., 2018), while benefiting from heuristics such as noise perturbed actions sampling and entropy maximization, do not sufficiently emphasize principled and directed exploration in their design principles. On the flip side, LSAC is highly exploratory in nature and effectively increases state-coverage during training by virtue of using theoretically principled LMC based Thompson sampling.

## 1.1 Key Contributions

To address the aforementioned challenges, we propose *Langevin Soft Actor-Critic* (LSAC) that endows traditional Max-Ent actor-critic algorithms with LMC-style updates and multimodal posterior sampling techniques for $Q$ function. We summarize our algorithmic contributions as follows:

**Distributional Adaptive Langevin Monte Carlo.**  Incorporating LMC for updating the $Q$ function significantly boosts exploration while simultaneously maintaining a crucial balance with exploitation. To address (C2), we first define a distributed Max-Ent critic objective inspired by Duan et al. (2023). Then, we employ distributional critic learning with the addition of adaptive LMC samplers.

**Multimodal $Q$ Posteriors.**  One downside of naive LMC updates is that potentially homogeneous posterior samples are generated at adjacent gradient steps. This translates to sampling similar critic for adjacent areas of $Q$-function posterior and this high correlation restricts exploration within the policy space while using a single critic for policy update. Consequently, a much longer burn-in period (Roy, 2020) is necessary to ensure adequate mixing of the Markov chain. However, this extended burn-in period conflicts with the frequent updates required by the agent in complex exploration tasks. To overcome this challenge, we introduce a simplified version of parallel tempering or replica exchange method (Geyer & Thompson, 1995; Chandra et al., 2019) that helps with exploring different modes of the $Q$-posterior more effectively. This in turn diversifies the actions sampled by the Max-Ent policy.

**Diffusion $Q$ Action Gradient.**  In the off-policy model-free RL setting, directly sampling from diffusion policies can be prohibitively expensive (Chen et al., 2024), often requiring tens to hundreds of iterative inference steps per action, particularly in the absence of pretraining with a diffusion behavior model. To address this challenge and resolve (C3), we explore an alternative approach to introduce diversity and multimodality without relying solely on policy learning. Our strategy incorporates diffusion synthetic data, as proposed by Lu et al. (2024), to enhance critic updates. By blending online data with synthetic trajectories, and refining actions within the diffusion synthetic buffer, we leverage the $Q$ action gradient to effectively constrain synthetic actions within the support set of optimal actions. This approach reduces computational costs and ensures that synthetic actions effectively contribute to the stability and quality of critic learning, resulting in more accurate and robust value estimates.

## 2 PRELIMINARY

**Markov Decision Process and Maximum Entropy RL.** We consider *Markov Decision Process* (MDP) defined as a tuple $(\mathcal{S}, \mathcal{A}, P_0, P, R, \gamma)$ where $\mathcal{S}$ is a continuous state space, $\mathcal{A}$ is a continuous action space, $P_0$ is the initial state distribution, $P : \mathcal{S} \times \mathcal{A} \to \mathcal{S}$ is the transition probability, $R : \mathcal{S} \times \mathcal{A} \to \Delta(\mathbb{R})$ is the reward distribution function and $\gamma \in (0, 1)$ is the discount factor. At each timestep $t$, the agent observes a state $s_t \in \mathcal{S}$ and takes an action $a_t \sim \pi(a_t \mid s_t) \in \mathcal{A}$ following policy $\pi$ and transitions to the next state $s_{t+1}$ according to $s_{t+1} \sim P(s_{t+1} \mid s_t, a_t)$ while receiving the reward $R(s_t, a_t)$. For simplicity, we use the notation $(s, a)$ and $(s', a')$ as the current and the next state-action pairs respectively. Furthermore, we adopt $r_t$ to denote $R(s_t, a_t)$ and use $\rho_\pi(s)$ and $\rho_\pi(s, a)$ to denote the state and state-action occupancy measure induced by policy $\pi$.

While standard RL aims to find a policy that maximizes the expected cumulative return, in this work, we consider maximum entropy RL (Ziebart, 2010; Haarnoja et al., 2017) in which the objective function is augmented with the entropy of a policy at each visited state $s_t$:

$$J_\pi = \sum_{i=0}^{\infty} \mathbb{E}_{(s_i, a_i) \sim \rho_\pi} \gamma^i [r_i + \alpha \mathcal{H}(\pi(\cdot \mid s_i))], \tag{1}$$

where $\mathcal{H}(\pi(\cdot \mid s)) := \mathbb{E}_{a \sim \pi(\cdot \mid s)}[-\log \pi(a \mid s)]$ is the policy entropy and $\alpha > 0$ is a temperature coefficient. We denote the entropy augmented cumulative return from $s_t$, also known as soft return, by $G_t = \sum_{i=t}^{\infty} \gamma^i [r_i - \alpha \log \pi(a_i \mid s_i)]$. The soft Q-value of policy $\pi$, which describes the expected soft return of policy $\pi$ upon taking action $a_t$ at state $s_t$, is defined as $Q^\pi(s_t, a_t) := r_t + \gamma \mathbb{E}[G_{t+1}]$, where the expectation is taken over trajectory distribution under policy $\pi$.

**Langevin Monte Carlo (LMC).** LMC is a popular sampling algorithm in machine learning that leverages Euler discretization method to approximate the continuous-time Langevin diffusion process (Welling & Teh, 2011). Langevin diffusion (Rossky et al., 1978; Roberts & Stramer, 2002) is a stochastic process that is defined by the stochastic differential equation (SDE) $dw_t = -\nabla L(w_t)dt + \sqrt{2}dB_t$, where $L : \mathbb{R}^n \to \mathbb{R}$ is a twice-differentiable function and $B_t$ is a standard Brownian motion in $\mathbb{R}^d$. Taking Euler-Murayama discretization of the SDE, we obtain the iterative update rule for LMC:

$$w_{t+1} = w_t - \eta \nabla L(w_t) + \sqrt{2\eta\beta^{-1}} \epsilon_t, \tag{2}$$

where $\eta$ is a fixed step size, $\beta$ is the inverse temperature and $\epsilon_t \sim \mathcal{N}(0, I_d)$. LMC update generates a Markov chain whose stationary distribution converges to a target distribution $p(x) \propto \exp(-\beta L(w))$ (Roberts & Tweedie, 1996). Intuitively, LMC can be thought as a version of gradient descent perturbed by Gaussian noise. Replacing the true gradient $\nabla L(w_k)$ with some stochastic gradient estimators results in the celebrated stochastic gradient Langevin dynamics (SGLD) algorithm (Welling & Teh, 2011).

**Diffusion Models.** Diffusion models (Ho et al., 2020; Sohl-Dickstein et al., 2015) are a class of generative models that were inspired by non-equilibrium thermodynamics and first used in image synthesis. They have recently emerged as a powerful framework for RL to enhance multimodal decision-making process (Wang et al., 2023; Hansen-Estruch et al., 2023). Given data marginally distributed as $q_0(x_0)$, we sample from it by first defining a stochastic differential equation (SDE)

$$dx_t = f(t)x_t dt + g(t)dw_t, \quad x_0 \sim q_0(x_0), \tag{3}$$

where $w_t$ is the standard $d$-dimensional Wiener process. Diffusion models gradually add Gaussian noise from $t = 0$ to $T$ by setting noise schedules $\sigma_{\max} = \sigma_T > \sigma_{T-1} > \cdots > \sigma_0 = 0$ such that $x_t \sim q_t(x_t; \sigma_t)$ and $q_T \sim \mathcal{N}(0, \sigma_{\max}^2 I)$ is indistinguishable from pure Gaussian noise. Equation 3 admits an equivalent reverse denoising process starting with the fully noised distribution $q_T(x_T)$:

$$dx_t = (f(t)x_t - g^2(t)\nabla_x \log q_t(x_t))dt + g(t)d\bar{w}_t, \quad x_T \sim q_T(x_T), \tag{4}$$

Since the score function $\nabla_x \log q_t(x_t)$ at each time step $t$ is unknown, Karras et al. (2022) considers training a noise predictor $\epsilon_\theta(x_t, t)$ on the score matching objective

$$L_{\text{VLB}}(\theta) = \min_\theta \mathbb{E}_{x \sim q_0(x_0), \epsilon \sim \mathcal{N}(0, \sigma^2 I)} \|D_\theta(x + \epsilon; \sigma) - x\|_2^2.$$

to predict the added noise $\epsilon$ that converts $x_0$ to $x_T$. Then, the score function can be expressed as $\nabla_x \log p_0(x; \sigma) = (D_\theta(s; \sigma) - x)/\sigma^2$, and we can generate synthetic samples by solving either the backward SDE in Equation 4 or using DPM solvers (Lu et al., 2022).

## 3 Algorithm Design

In this section, we introduce *Langevin Soft Actor Critic* (LSAC), as shown in Algorithm 1, which builds off three main ideas. First, during critic learning, we want to learn and efficiently sample a Q-value function from its approximate posterior distribution. We leverage Langevin Monte Carlo (LMC) to perform this. This is a natural adaptation to posterior sampling or Thompson sampling that is widely used in RL for efficient exploration. Second, we couple LMC based posterior sampling with distributional value function learning (Duan et al., 2023; Ma et al., 2020) that helps with mitigating the well-known overestimation issue. Third, to ensure LMC can sample from different modes of Q-posterior, we use parallel tempering (Chandra et al., 2019). Fourth, to improve sample efficiency and the UTD ratio, during critic update, we synthesize diverse and potentially high-valued state-action samples using a diffusion model and $Q$ action gradient refinement.

---

**Algorithm 1:** Langevin Soft Actor-Critic (LSAC)

**Input:** Policy networks $\pi_\phi, \pi_{\bar\phi}$, critic networks $\mathcal{Z}_\psi, \mathcal{Z}_{\bar\psi}$, and diffusion model $\mathcal{M}$.
Replay buffer $\mathcal{D} \leftarrow \emptyset$, diffusion buffer $\mathcal{D}' \leftarrow \emptyset$.
Collection of posteriors $\Psi_{\mathcal{Z}} = \{\psi^{(i)}\}_{i=1}^n$, entropy factor $\alpha$, initialize weights $\psi^{(i)}$ for $\{\psi^{(i)}\}_{i=1}^n$.
Set step size $\eta_Q > 0$ and temperatures $\beta_a, \beta_\alpha, \beta_\pi, \beta_Q, \beta_M$.

1  **while** *policy has not converged* **do**
2     **for** *each sampling step* **do**
3         ⌊ Online interaction with $\pi_\phi$ in the environment, $\mathcal{D} \leftarrow \mathcal{D} \cup \{(s, a, r, s', d)\}$.
4     **for** *each update step* **do**
5         **for** $i = 1, \ldots, n$ **do**
6             Sample mini-batch $B_{D_i}$ from $\mathcal{D}$ and $B_{M_i} = \{(s_M, a_M, r_M, s'_M, d_M)\}$ from $\mathcal{D}'$.
7             Refine $a_M$ with $a_M \leftarrow a_M + \beta_a \nabla_a Q_{\psi^{(i)}}(s_M, a_M)$ in $B_{M_i}$.
8             Update $\mathcal{Z}_{\psi^{(i)}}$ on $B_i = B_{D_i} \cup B_{M_i}$ with Algorithm 2.
9         Sample $\psi^{(i)} \sim \mathcal{U}(\Psi_{\mathcal{Z}})$ at random and recover $B_i$.
10        Compute $\alpha$ with Equation 12 and update $\pi_\phi$ on $B_i$ with Equation 11.
11        Update $\mathcal{M}$ on $\mathcal{D}$ with Equation 10 and fill up $\mathcal{D}'$.
12        Polyak update $\bar\psi^{(i)} \leftarrow \tau\psi^{(i)} + (1-\tau)\bar\psi^{(i)}$, $\bar\phi \leftarrow \tau\phi + (1-\tau)\bar\phi$.

---

**Distributional Critic Learning with Adaptive Langevin Monte Carlo.**  To describe distributional critic update, we first define few terminologies. We first define soft state-action return, a random variable, given by $Z^\pi(s_t, a_t) := r_t + \gamma G_{t+1}$, which is a function of policy $\pi$ and state-action pair $(s_t, a_t)$. It is easy to observe that $Q^\pi(s, a) = \mathbb{E}[Z^\pi(s, a)]$. Instead of the expected state-action return $Q^\pi(s, a)$, we aim to model the distribution of the random variable $Z^\pi(s, a)$. We define $\mathcal{Z}^\pi(Z^\pi(s, a) \mid s, a) : \mathcal{S} \times \mathcal{A} \to \mathcal{P}(Z^\pi(s, a))$ as a mapping from $(s, a)$ to a distribution over the soft state-action return $Z^\pi(s, a)$. We refer to this mapping as value distribution function. We define the distributional Bellman operator in the maximum entropy framework as

$$\mathcal{T}^\pi Z^\pi(s, a) \overset{D}{:=} r + \gamma(Z^\pi(s', a') - \alpha \log \pi(a' \mid s')). \tag{5}$$

We model the value distribution function and stochastic policy as diagonal Gaussian distribution and parameterize as $\mathcal{Z}_\psi(\cdot \mid s, a)$ and $\pi_\phi(\cdot \mid s)$, where $\psi$ and $\phi$ are the neural network parameters. Due to Gaussian assumption, $\mathcal{Z}_\psi$ can be expressed as $\mathcal{Z}_\psi(\cdot \mid s, a) = \mathcal{N}(Q_\psi(s, a), \sigma_\psi(s, a)^2)$, where $Q_\psi(s, a)$ and $\sigma_\psi(s, a)$ are the mean and standard deviation of value distribution respectively. The distributional critic is updated by minimizing the following loss function:

$$L_{\mathcal{Z}}(\psi) := \omega \mathbb{E}_{(s,a)\sim B} D_{\mathrm{KL}}(\mathcal{T}^{\pi_{\bar\phi}} \mathcal{Z}_{\bar\psi}(s, a) \| \mathcal{Z}_\psi(s, a)), \tag{6}$$

where $D_{\mathrm{KL}}$ is the Kullback-Leibler (KL) divergence, $B$ is the replay buffer, and $\bar\psi$ and $\bar\phi$ denotes the target network parameters of $\mathcal{Z}_\psi$ and $\pi_\phi$ respectively. The gradient scalar $\omega := \mathbb{E}_{(s,a)\sim B}[\sigma_\psi(s, a)^2]$ depends on the variance of the distribution function.

Following Duan et al. (2023), we decompose the critic update gradient into two components: mean-related gradient $\nabla_\psi L_{\mathcal{Z},m}(\psi)$ and variance-related gradient $\nabla_\psi L_{\mathcal{Z},v}(\psi)$:

$$\nabla_\psi L_{\mathcal{Z},m}(\psi) := -\frac{y_Q - Q_\psi(s,a)}{\sigma_\psi(s,a)^2 + \epsilon_\sigma} \nabla_\psi Q_\psi(s,a)$$

$$\nabla_\psi L_{\mathcal{Z},v}(\psi) := -\frac{(\mathrm{clip}_b(y_Z) - Q_\psi(s,a))^2 - \sigma_\psi(s,a)^2}{\sigma_\psi(s,a)^3 + \epsilon_\sigma} \nabla_\psi \sigma_\psi(s,a),$$

where the target terms $y_Q$ and $y_Z$ are defined as $y_Q := r + \gamma(Q_{\bar\psi}(s',a') - \alpha \log \pi_{\bar\phi}(a'|s'))$ and $y_Z := r + \gamma(Z_{\bar\psi}(s',a') - \alpha \log \pi_{\bar\phi}(a'|s'))$. $y_Z$ is further clipped with a clipping function $\mathrm{clip}_b(y_Z) := \mathrm{clip}(y_Z, Q_\psi(s,a) - b, Q_\psi(s,a) + b)$ where $b \propto \mathbb{E}_{(s,a)\sim B}\sigma_\psi(s,a)$ is an automated boundary.

Finally, we can express the sample-based critic update gradient as

$$\nabla_\psi L_{\mathcal{Z}}(\psi) \approx \mathbb{E}_{(s,a)\sim B}\left[\nabla_\psi L_{\mathcal{Z},m}(\psi) + \nabla_\psi L_{\mathcal{Z},v}(\psi)\right], \tag{7}$$

In Appendix B, we show that, under some mild assumptions, the posterior over $Q_\psi$ is of the form $\exp(-L_{\mathcal{Z}}(\psi))/Z$, where $Z$ is the partition function, and that $Q_\psi(s,a) = \mathbb{E}_{(s,a)\sim B}\mathcal{Z}_\psi(s,a)$. However, exactly sampling from this distribution is non-trivial as we do not know the partition function. To this mean, we can use LMC based sampling algorithm. In place of vanilla LMC described in Equation 2, following Ishfaq et al. (2024a); Kim et al. (2022), we use adaptive Stochastic Gradient Langevin Dynamics (aSGLD), where an adaptively adjusted bias term is included in the drift function to enhance escape from saddle points and accelerate the convergence to the true $Q$ posterior, even in the presence of pathological curvatures and saddle points which are common in deep neural network (Dauphin et al., 2014). Concretely, we use the following update rule

$$\psi_{k+1} \leftarrow \psi_k - \eta_Q(\nabla_\psi L_{\mathcal{Z}}(\psi_k) + a\zeta_{\psi_k}) + \sqrt{2\eta_Q \beta_Q^{-1}}\epsilon_k, \quad \epsilon_k \sim \mathcal{N}(0, I_d), \tag{8}$$

for a step size $\eta_Q > 0$, bias factor $a$, adaptive preconditioner $\zeta_k$, and inverse temperature $\beta_Q$. Inspired from the Adam optimizer (Kingma & Ba, 2014), the adaptive preconditioner $\zeta_k$ is defined as $\zeta_{\psi_k} := m_k \oslash \sqrt{v_k + \lambda\mathbf{1}}$ where,

$$m_k = \alpha_1 m_{k-1} + (1-\alpha_1)\nabla L_{\mathcal{Z}}(\psi_k) \quad \text{and} \quad v_k = \alpha_2 v_{k-1} + (1-\alpha_2)\nabla L_{\mathcal{Z}}(\psi_k) \odot \nabla L_{\mathcal{Z}}(\psi_k), \tag{9}$$

with $\alpha_1, \alpha_2 \in [0,1)$ being the smoothing factors of the first and second moments of the stochastic gradients, respectively. Each sampled $\psi$ following Equation 8 parameterizes a possible distributional $Q$ function and thus is equivalent to sampling from the posterior over distributional $Q$ function.

While our critic update rule is motivated by Ishfaq et al. (2024a), we are the first to apply aSGLD based parameter sampling for continuous control task along with distributional critic. Moreover, while in each critic update step, we perform one aSGLD update, Ishfaq et al. (2024a), in their LMCDQN algorithm, performs this update $\widetilde{O}(K)$ times, where $K$ is the episode number. This can significantly increase the runtime of LMCDQN compared to that of LSAC.

---

**Algorithm 2:** Distributional Adaptive Langevin Monte Carlo

**Input:** Policy $\pi_\phi$ and target $\pi_{\bar\phi}$, critic weight $\psi$ and target $\bar\psi$, data batch $B$.

1 Sample $\epsilon \sim \mathcal{N}(0, I_d)$.
2 Update $Z_\psi, Z_{\bar\psi}$ and $\mathcal{T}^{\pi_{\bar\phi}}Z_{\bar\psi}(s,a)$ by Equation 5.
3 Set clipping boundary $b \propto \mathbb{E}_{(s,a)\sim B}\sigma_\psi(s,a)$, temperature $\omega \propto \mathbb{E}_{(s,a)\sim B}[\sigma_\psi(s,a)^2]$.
4 Set $L_{\mathcal{Z}}(\psi) := \omega\mathbb{E}_{(s,a)\sim B}D_{\mathrm{KL}}(\mathcal{T}^{\pi_{\bar\phi}}\mathcal{Z}_{\bar\psi}(s,a)\|\mathcal{Z}_\psi(s,a))$ by Equation 6.
5 Compute the adaptive drift bias $\zeta_\psi := m \oslash \sqrt{v + \lambda\mathbf{1}}$ using Equation 9.
6 Update $\mathcal{Z}$ posterior weights with $\psi \leftarrow \psi - \eta_Q(\nabla_\psi L_{\mathcal{Z}}(\psi) + a\zeta_\psi) + \sqrt{2\eta_Q\beta_Q^{-1}}\epsilon$.
7 Polyak update $b \leftarrow \tau b + (1-\tau)\mathbb{E}_{(s,a)\sim B}[\sigma_\psi(s,a)], \omega \leftarrow \tau\omega + (1-\tau)\mathbb{E}_{(s,a)\sim B}[\sigma_\psi(s,a)^2]$.

---

**Parallel Tempering and Multimodal $Q$ Posteriors.** Despite the scalability of LMC, its mixing rate is often extremely slow, especially for distributions with complex energy landscapes (Li et al., 2018). Performing naive LMC to approximately sample from multimodal $Q$ posterior can thus converge very slowly which in turn will affect the performance of the algorithm. Parallel tempering (also known

as replica exchange) (Marinari & Parisi, 1992; Geyer & Thompson, 1995; Chandra et al., 2019) is a standard approach for exploring multiple modes of the posterior distribution while performing LMC. In parallel tempering, multiple MCMC chains (known as replicas) are executed at different temperature values. It allows global and local exploration which makes it suitable for sampling from multi-modal distributions (Hukushima & Nemoto, 1996; Patriksson & van der Spoel, 2008).

We use a simplified version of parallel tempering where for all replicas we use the same temperature. To reduce complexity, we also do not perform replica exchange. Even though, in principle, it can limit the exploration of the parameter space, as we initialize each replica with different starting points, it achieves enough exploration for our purpose while maintaining a simple implementation. By running multiple LMC chains $\Psi_Q = \{\psi^{(i)}\}_{i=1}^n$, we can sample $Q$-functions for critics from distinct modes of the multimodal posterior distribution while ensuring faster convergence and mixing time.

**Diffusion $Q$ Action Gradient.** Our approach begins with $\pi_\phi$, which approximates a Max-Ent policy (Eysenbach & Levine, 2022) used for collecting online trajectories $\tau$. To enhance the diversity of state-action pairs and increase the UTD ratio for critic updates, we first randomly sample a batch $B_{D_i}$ from the online buffer. Next, we sample synthetic data $B_{M_i}$ from a diffusion (Wang et al., 2023; Lu et al., 2024) generator $\mathcal{M}$.

However, between the periodic updates of $\mathcal{M}$, synthetic data generated by $\mathcal{M}$ can become stale, potentially limiting its effectiveness in dynamic environments. Hence, each action sample in $B_{M_i}$ is then refined through gradient ascent $\beta_a \nabla_a Q_{\psi^{(i)}}(s, \widetilde{a})$, targeting improved alignment with high-value regions. While this action gradient approach shares conceptual similarities with DIPO (Yang et al., 2023), which replaces the original actions from the samples in the replay buffer by performing gradient ascent for policy optimization, our dual focus on diversity and quality of mini-batch data used for critic and policy update is distinct. We utilize $Q$ action gradient specifically for critic updates, ensuring that the synthetic actions are not only diverse but also accurately reflect regions of high $Q$ value, all while remaining within the valid support set of the action space. Finally, to increment the UTD ratio, we mix $B_{D_i}$ and $B_{M_i}$ into a single parallel data batch $B_i$ and for each $1 \leq i \leq n$, update $\psi^{(i)}$ using Algorithm 2.

The diffusion model $\mathcal{M}$ is trained with the score matching loss

$$L_{\mathcal{M}}(\theta) := \mathbb{E}_{t \sim \mathcal{U}([T]), z \sim \mathcal{N}(0, I_d), (s,a) \sim \mathcal{D}} \| z - \epsilon_\theta(\sqrt{\bar{\alpha}_t} a + \sqrt{1 - \bar{\alpha}_t} z, s, t) \|_2^2, \qquad (10)$$

where $\mathcal{U}([T])$ denotes the uniform distribution over a finite collection of reverse time indices $\{1, \ldots, T\}$. The weights $\bar{\alpha}_t := \prod_{i=1}^t \alpha_i$, $\alpha_i := 1 - \beta_i$, are computed via predefined diffusion temperatures $\{\beta_i\}_{i=1}^T$.

**Policy Improvement.** For each actor update step, we randomly sample a $\mathcal{Z}$ weight $\psi^{(i)}$ from $\Psi_{\mathcal{Z}}$ and retrieve the mixed replay data batch $B_i$. Soft policy improvement maximizes the usual Max-Ent objective

$$\begin{aligned} L_\pi(\phi) &= \mathbb{E}_{s \sim B_i, a \sim \pi_\phi}[Q_{\psi^{(i)}}(s, a) + \alpha \mathcal{H}(\pi_\phi(a|s))] \\ &= \mathbb{E}_{s \sim B_i, a \sim \pi_\phi} \Big[ \mathbb{E}_{Z(s,a) \sim \mathcal{Z}_{\psi^{(i)}}(\cdot \mid s, a)}[Z(s, a)] + \alpha \mathcal{H}(\pi_\phi(a|s)) \Big]. \end{aligned} \qquad (11)$$

Following Haarnoja et al. (2018a;b), the entropy coefficient $\alpha$ is updated with

$$\alpha \leftarrow \alpha - \beta_\alpha \nabla_\alpha(-\log \pi_\phi(a|s) - \overline{\mathcal{H}}), \qquad (12)$$

where $\overline{\mathcal{H}}$ is the expected entropy. Finally, we update the diffusion model $\mathcal{M}$ periodically with on-policy data and generate $|\mathcal{D}'|$ copies of synthetic transitions into $\mathcal{D}'$, while target networks are updated using the Polyak averaging approach.

# 4 EXPERIMENTS

## 4.1 EXPERIMENTS IN MUJOCO AND DMC

**Main Results.** We present empirical evaluations of LSAC on the MuJoCo benchmark (Todorov et al., 2012; Brockman et al., 2016) and the DeepMind Control Suite (DMC) (Tassa et al., 2018),

showing that LSAC is able to outperform or match several strong baselines, including DSAC-T (Duan et al., 2023), the current state-of-the-art model-free off-policy RL algorithm. Other baselines include DIPO (Yang et al., 2023), SAC (Haarnoja et al., 2018a), TD3 (Fujimoto et al., 2018), PPO (Schulman et al., 2017), TRPO (Schulman et al., 2015), REDQ (Chen et al., 2021) and QSM (Psenka et al., 2024). Our code is available at `https://github.com/hmishfaq/LSAC`.

We emphasize that, for implementation simplicity and fair comparisons, both policy and critic networks sizes are kept the same for our algorithm and all of the baselines. After an initial warm-up stage of `1e5` steps, we gradually anneal LMC step size $\eta_Q$ from the initial `1e-3` down to `1e-4`. For computing the adaptive drift bias $\zeta_\psi$, we use fixed values of $\alpha_1 = 0.9$, $\alpha_2 = 0.999$ in Equation 9, and $\lambda = 10^{-8}$ without tuning them. To prevent gradient explosion during training, we clip the sum of the gradient and the adaptive bias term using $\text{clip}_c(\nabla_\psi L_Q(\psi) + a\zeta_\psi)$ by a constant $c = 0.7$.

We accelerate training following the SynthER (Lu et al., 2024) implementation and update the diffusion generator $\mathcal{M}$ using the data $\mathcal{D}$ every `1e4` time steps. During the critic updates, for each $\psi^{(i)} \in \Psi_Q$, where $1 \le i \le n$, the sampled replay buffer data is mixed with a synthetic batch $B_{M_i}$ with a ratio of 0.5. This synthetic batch is generated from the diffuser, with its state-action samples immediately optimized through gradient ascent with respect to the $Q$ function, parameterized by the current weight $\psi^{(i)}$.

From Figure 1 and Table 1, we see that LSAC outperforms other baselines in 5 out of 6 tasks from MuJoCo. In Humanoid-v3 even though DSAC-T outperforms LSAC, the difference is marginal. For space constraint we report the DMC result in Appendix C and Figure 8.

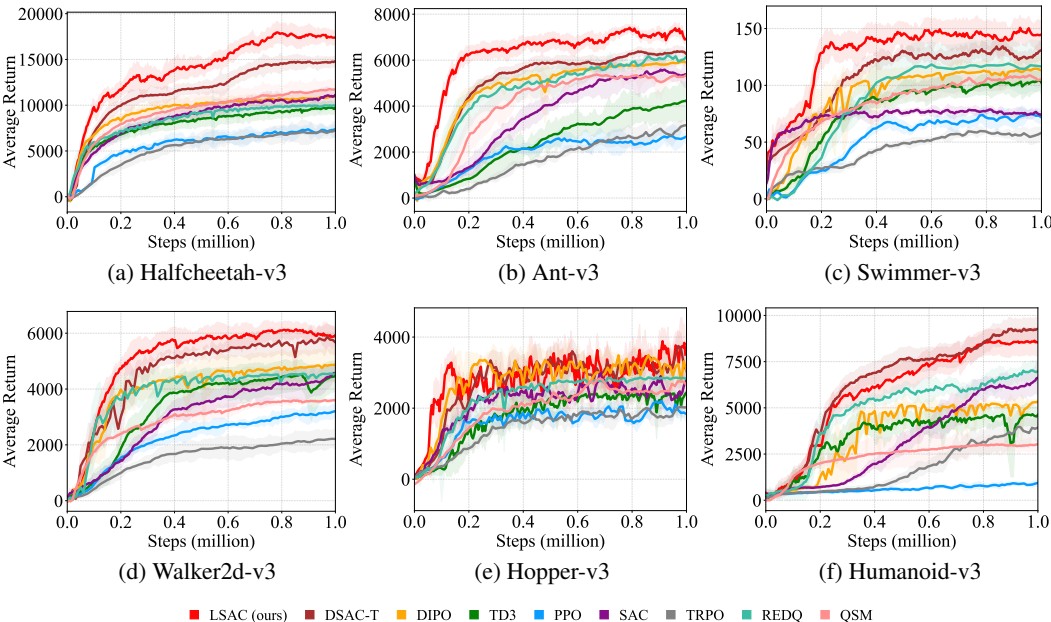

Figure 1: Training curves for six MuJoCo continuous control tasks over `1e6` time steps. Results are averaged over a window size of 11 epochs and across 10 seeds. Solid lines represent the median performance, and the shaded regions correspond to 90% confidence interval.

| Environments\Methods | LSAC (ours) | DSAC-T | DIPO | SAC | TD3 | PPO | TRPO | REDQ |
|---|---|---|---|---|---|---|---|---|
| HalfCheetah | **17948 ± 1724** | 12703 ± 1711 | 9329 ± 1798 | 10543 ± 1422 | 9034 ± 1350 | 6560 ± 1189 | 6534 ± 1345 | 10022 ± 1298 |
| Ant | **7411 ± 155** | 6153 ± 211 | 5459 ± 163 | 5297 ± 289 | 4839 ± 271 | 3055 ± 131 | 3271 ± 146 | 6091 ± 129 |
| Swimmer | **151 ± 11** | 129 ± 9 | 114 ± 11 | 76 ± 5 | 102 ± 10 | 76 ± 6 | 60 ± 4 | 134 ± 22 |
| Walker2d | **6143 ± 394** | 5880 ± 411 | 4921 ± 549 | 4535 ± 402 | 4625 ± 399 | 3182 ± 233 | 2228 ± 302 | 4598 ± 318 |
| Hopper | **3839 ± 537** | 3327 ± 588 | 3138 ± 731 | 2919 ± 165 | 2604 ± 140 | 2315 ± 152 | 2096 ± 201 | 3002 ± 512 |
| Humanoid | 8545 ± 740 | **9028 ± 792** | 5012 ± 811 | 6807 ± 734 | 4455 ± 820 | 1018 ± 102 | 4459 ± 564 | 7213 ± 621 |

Table 1: Maximum Average Return across 10 seeds over `1e6` time steps. Maximum value and corresponding 90% confidence interval for each task are shown in bold.

**Sensitivity Analysis.** In Figure 2, we present the learning curves of LSAC for different values of learning rates $\eta_Q \in \{10^{-2}, 10^{-3}, 3 \times 10^{-4}, 10^{-4}\}$, inverse temperature $\beta_Q \in \{10^5, 10^6, 10^7, 10^8, 10^9\}$, and bias factor $a \in \{10, 1, 0.1, 0.01\}$. We observe that our algorithm is most sensitive to the step size $\eta_Q$ in the LMC update and the bias factor $a$ from Equation 8. On the contrary, LSAC is less sensitive to the choice of the inverse temperature $\beta_Q$.

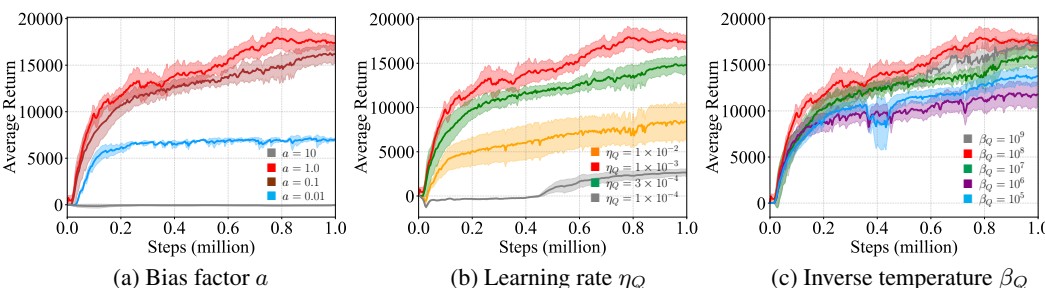

(a) Bias factor $a$     (b) Learning rate $\eta_Q$     (c) Inverse temperature $\beta_Q$

Figure 2: Sensitivity analysis of different parameters on HalfCheetah-v3 environment. A comparison of LSAC with different bias factors $a$, step sizes $\eta_Q$, and inverse temperature parameters $\beta_Q$.

We now present a comprehensive ablation analysis of LSAC by systematically removing individual algorithmic contributions while maintaining optimal parameters for the remaining components. We refer the readers to Table 4 for a complete list of hyperparameters used for each model.

**Impact of distributional critic on performance and overestimation bias.** To understand the impact of distributional critic, we run ablation studies where we replace our distributional critic with a standard critic implementation in SAC (Haarnoja et al., 2018a). Figure 3 shows that the performance of LSAC declines significantly when distributional critic is replaced by standard critic. To find out what might be driving such performance gap, following the same evaluation protocol as Chen et al. (2021), we compare the normalized $Q$ estimation biases in Figure 4. We observe that throughout most of training, LSAC with distributional critic has a much smaller and often near-constant underestimation bias compared to LSAC with standard critic. It indicates that distributional critic allows more stable learning and increased performance by lowering $Q$ estimation bias.

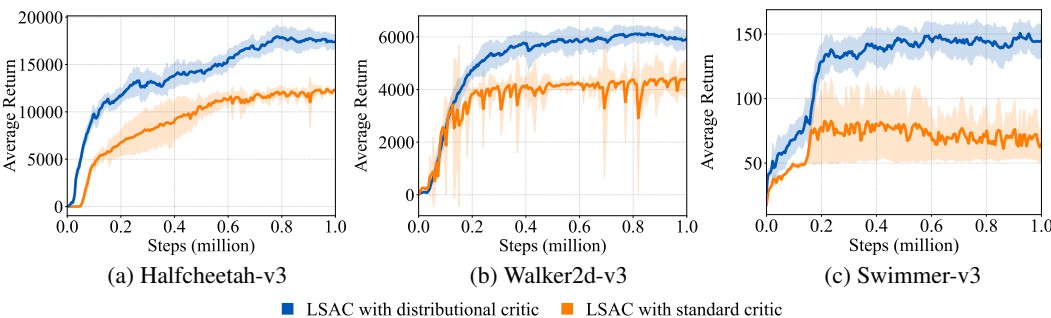

(a) Halfcheetah-v3     (b) Walker2d-v3     (c) Swimmer-v3

■ LSAC with distributional critic     ■ LSAC with standard critic

Figure 3: Ablation on MuJoCo environments comparing the replacement of the distributional critic component with a standard critic. LSAC with distributional critic is more performant than the variant where standard critic is used.

**Usefulness of the synthetic experience replay and action gradient ascent.** Figure 5a indicates that the performance of LSAC experiences only a marginal decline in HalfCheetah-v3 when the diffusion $Q$ action gradient is excluded. However, a more pronounced drop is observed in Ant-v3 (Figure 5b) and Swimmer (Figure 5c) when action gradient is excluded.

**Number of parallel critics.** In Figure 6a, we observe that when LSAC is equipped with too few or too many parallel critics $|\Psi_Q|$, the performance drops. This is due to when the parallel critic number

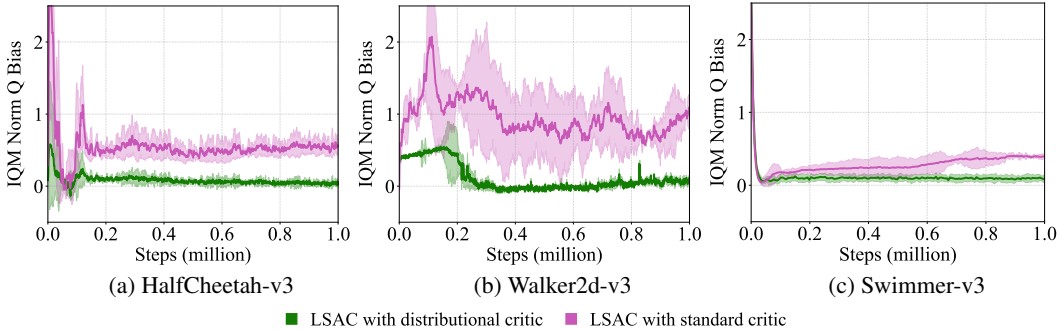

Figure 4: Normalized $Q$ bias plots for ablation study of the distributional critic component in LSAC. The $Q$ bias value is estimated using the Monte Carlo return over 1e3 episodes on-policy, starting from states sampled in the replay buffer.

is too low, the LMC sampler cannot explore different modes of the posterior distribution. On the other hand, when the critic number is high, it may hamper the actor learning as during each policy update it may encounter some critics only very few times due to uniform sampling of the critic. This may cause drop in the performance.

**Usefulness of aSGLD sampler.** In Figure 6b, we observe that approximate Thompson sampling through aSGLD sampler boosts the performance compared to when the critics are simply trained with the Adam (Kingma & Ba, 2014) optimizer. When only Adam is used, the collection of critics can be thought of as an ensemble akin to bootstrapped DQN (Osband et al., 2016a).

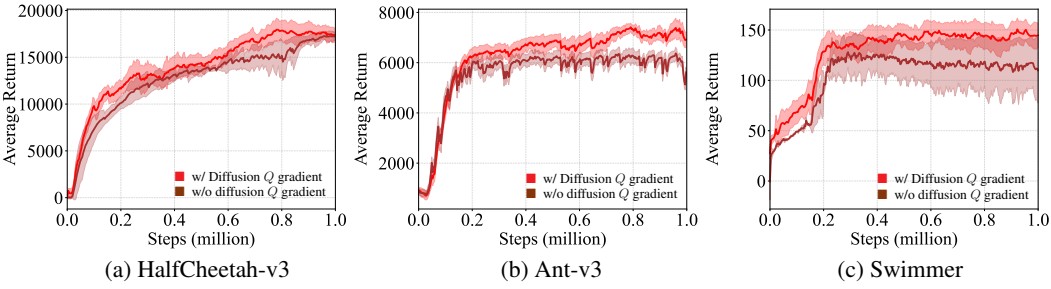

Figure 5: Ablation study of $Q$ action gradient regularization of synthetic state-action samples on the effect of average return in three MuJoCo environments.

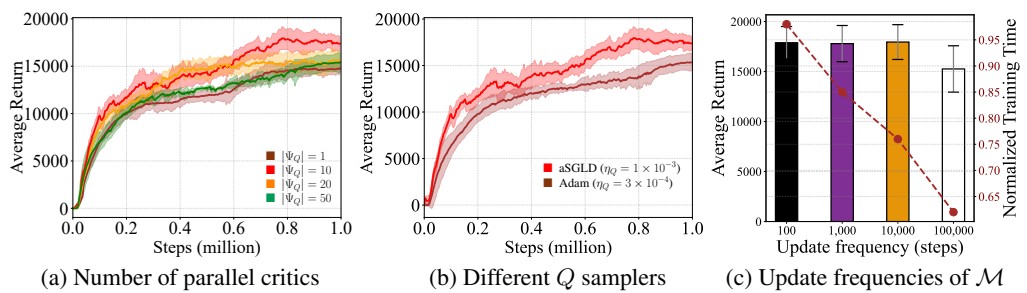

Figure 6: Ablation study on HalfCheetah-v3 environment. Performance of LSAC is affected by (a) the choice of parallel critics number and (b) the use of LMC (aSGLD) sampler. (c) The performance difference between each update frequency of diffusion model $\mathcal{M}$ is not significant.

**Learning is stable in practice.** While off-policy deep RL algorithms are often challenging to stabilize, we found that LSAC is fairly stable as shown in Figure 18. This is likely due to the KL objective on which parallel distributional critics are optimized, where the stochastic soft state-action value $Z_\psi$ remains close to the value target distribution $\mathcal{T}^{\pi_{\bar{\phi}}} Z_{\bar{\psi}}$. Moreover, distributional critic stabilizes learning by mitigating overestimation bias.

## 4.2 EXPLORATION CAPABILITY OF LSAC

To further evaluate the exploration ability of LSAC, we test our method on two types of maze environments, a custom version of `PointMaze_Medium-v3` and `AntMaze-v4` from de Lazcano et al. (2024), which are implemented based on the D4RL benchmark (Fu et al., 2020). In `PointMaze_Medium-v3`, the agent is tasked with manipulating a ball to reach some unknown goal position in the maze. The initial state of the ball is at the center of the maze and we define two potential goal states for the ball – the top right and the bottom left corner of the maze. Please refer to Appendix D for further details on the environments. We first train the agent for $500k$ environment steps, and then use its oracle to complete 200 evaluation episodes. The agent has better exploration ability if it solves the task by reaching multiple goals or finding out multiple paths leading to a goal.

To quantify the exploration ability of LSAC and baseline methods, we discretize the maze and track the cell visitation to visualize the exploration density map and track the cell visitation. We set the maximum density threshold to be 100 visits per cell to reduce the dominance of high-density areas such as the agent's start location, which may otherwise interfere with measuring the true trajectory densities. In Figure 7 and Figure 9, we see that LSAC is capable of discovering multiple paths leading to both goals while all other baselines, except for DIPO (Yang et al., 2023), either fail to solve the task or only manage to discover a single path. While DIPO manages to find multiple paths toward the goal, LSAC offers state coverage that is comparable to or greater than that of DIPO, as shown in Figure 10.

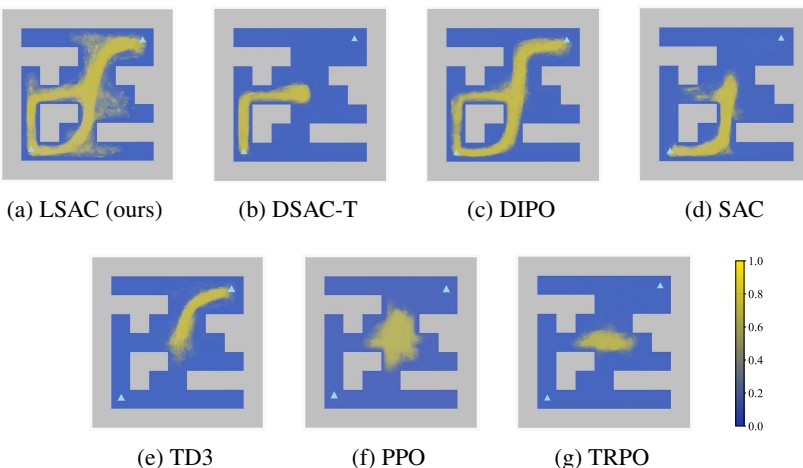

(a) LSAC (ours)    (b) DSAC-T    (c) DIPO    (d) SAC

(e) TD3    (f) PPO    (g) TRPO

Figure 7: Exploration density maps of LSAC and baseline algorithms tested on the `PointMaze_Medium-v3` environment. The two goals are located in the upper-right and lower-left corners, as shown by the triangle markers. The starting position is at the center of the maze map.

## 5 CONCLUSION

In this paper, we introduced LSAC, an off-policy algorithm that leverages LMC based approximate Thompson sampling to learn distributional critic. We observe that distributional critic learning coupled with LMC based exploration can boost performance while mitigating overestimation issue commonly seen in other model-free methods. Future work includes trying more advanced approximate samplers such as underdamped Langevin Monte Carlo (Ishfaq et al., 2024b).

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

## A    RELATED WORK

**Posterior sampling.**    Our research is closely aligned with approaches that utilize posterior sampling, specifically Thompson sampling, within the reinforcement learning (RL) framework (Strens, 2000). Notably, Osband et al. (2016b), Russo (2019), and Xiong et al. (2022) introduced randomized least-squares value iteration (RLSVI), which incorporates frequentist regret analysis in the context of tabular MDPs. RLSVI strategically adds carefully calibrated random noise to the value function to promote exploration. Building on this, Zanette et al. (2020) and Ishfaq et al. (2021) extended RLSVI to linear MDP settings. Although RLSVI achieves favorable regret bounds in both tabular and linear scenarios, its reliance on predefined and fixed features during training limits its applicability to deep RL environments (Li et al., 2021).

To address this limitation, Osband et al. (2016a; 2018) proposed training an ensemble of randomly initialized neural networks, treating them as approximate posterior samples of Q functions. However, this ensemble approach incurs significant computational overhead. Alternatively, some studies have explored directly injecting noise into network parameters (Fortunato et al., 2018; Plappert et al., 2018). For instance, Noisy-Net (Fortunato et al., 2018) learns noisy parameters through gradient descent, while Plappert et al. (2018) introduced constant Gaussian noise to the network parameters. Nonetheless, Noisy-Net does not guarantee an accurate approximation of the posterior distribution (Fortunato et al., 2018).

Dwaracherla & Van Roy (2020); Ishfaq et al. (2024a;b) propose using Langevin Monte Carlo for approximate Thompson sampling which is the most related work to ours. Furthermore, Ciosek et al. (2019) explored bootstrapped DQN-inspired actor-critic algorithms but were unable to manage scenarios involving multimodal Q posterior distributions.

**Upsampling in RL training.**    Prior RL studies that augment existing datasets typically employ Generative Adversarial Networks (GANs) (Goodfellow et al., 2014) or Variational Auto-Encoders (VAEs) (Kingma et al., 2019). For example, Huang et al. (2017) utilized GANs to generate synthetic data for pre-training RL agents, thereby accelerating training in production environments. Similarly, Lee et al. (2020) applied sequential latent variable models like VAEs to perform amortized variational inference in partially observable Markov Decision Processes (POMDPs). However, as highlighted by Lu et al. (2024), these methods often face limitations in achieving rapid training in online proprioceptive settings and scalability in data synthesis.

**Online reinforcement learning with diffusion.**    Recently, there has been growing interest in using diffusion model to represent policies in online reinforcement learning due to its inherent ability in learning complex and multimodal distributions. One of the earliest works, that employ diffusion policies for online RL is DIPO (Yang et al., 2023). DIPO uses the critic to update the sampled action from the replay buffer using action gradient before fitting the actor using the updated actions from the replay buffer. Psenka et al. (2024) argues that optimizing the likelihood of the entire chain of denoised actions can be computationally inefficient and instead proposes Q-Score Matching (QSM) that iteratively aligns the gradient of the diffusion actor (i.e. score) with the action gradient of the critic. Li et al. (2024) proposes DDiffPG — an actor-critic algorithm that learns multimodal policies parameterized as diffusion models from scratch. To discover different modes in the policy, DDiffPG uses novelty-based intrinsic motivation along with off-the-shelf unsupervised hierarchical clustering methods. More recently, Wang et al. (2024) proposes DACER that uses the reverse process of the diffusion model as a policy function. To perform adaptive adjustment of the exploration level of the diffusion policy, DACER estimates the entropy of the diffusion policy using Gaussian mixture model. We emphasize that while these works utilize diffusion models to parameterize policies, we use diffusion model to create synthetic data to enhance critic learning.

## B    THEORETICAL INSIGHTS

Without considering the gradient scalar $\omega$, the objective function of the critic update from Equation 6 can be written as

$$L_{\mathcal{Z}}(\psi) = \mathbb{E}_{(s,a)\sim B} D_{\mathrm{KL}}(\mathcal{T}^{\pi_{\bar{\phi}}} \mathcal{Z}_{\bar{\psi}}(s,a) \| \mathcal{Z}_{\psi}(s,a)). \tag{13}$$

Using Proposition B.1, it can be further shown that the objective function in Equation 13 is equivalent to the following:

$$L_{\mathcal{Z}}(\psi) = -\mathbb{E}_{\substack{(s,a,r,s')\sim B, a'\sim\pi_{\bar{\phi}}, \\ Z(s',a')\sim\mathcal{Z}_{\bar{\psi}}(\cdot\,|\,s',a')}} \big[\log\mathbb{P}(\mathcal{T}^{\pi_{\bar{\phi}}}Z(s,a)\,|\,\mathcal{Z}_{\psi}(\cdot\,|\,s,a))\big] + c, \tag{14}$$

where $c$ is a term independent of $\psi$.

Since $\mathcal{Z}_{\psi}$ is assumed to be a Gaussian model, it can be expressed as $\mathcal{Z}_{\psi}(\cdot\,|\,s,a) = \mathcal{N}(Q_{\psi}(s,a), \sigma_{\psi}(s,a)^2)$, where $Q_{\psi}(s,a)$ and $\sigma_{\psi}(s,a)$ are the outputs of the value network. Then ignoring the $\psi$ independent term $c$, Equation 14 can be written as

$$
\begin{aligned}
L_{\mathcal{Z}}(\psi) &= -\mathbb{E}_{\substack{(s,a,r,s')\sim B, a'\sim\pi_{\bar{\phi}}, \\ Z(s',a')\sim\mathcal{Z}_{\bar{\psi}}(\cdot\,|\,s',a')}} \log\left(\frac{\exp\left(-\frac{(\mathcal{T}^{\pi_{\bar{\phi}}}Z(s,a)-Q_{\psi}(s,a))^2}{2\sigma_{\psi}(s,a)^2}\right)}{\sqrt{2\pi}\sigma_{\psi}(s,a)}\right) \\
&= \mathbb{E}_{\substack{(s,a,r,s')\sim B, a'\sim\pi_{\bar{\phi}}, \\ Z(s',a')\sim\mathcal{Z}_{\bar{\psi}}(\cdot\,|\,s',a')}} \left[\frac{(\mathcal{T}^{\pi_{\bar{\phi}}}Z(s,a)-Q_{\psi}(s,a))^2}{2\sigma_{\psi}(s,a)^2} + \log(\sqrt{2\pi}\sigma_{\psi}(s,a))\right] \\
&= \mathbb{E}_{\substack{(s,a,r,s')\sim B, a'\sim\pi_{\bar{\phi}}, \\ Z(s',a')\sim\mathcal{Z}_{\bar{\psi}}(\cdot\,|\,s',a')}} \left[\frac{(y_Z-Q_{\psi}(s,a))^2}{2\sigma_{\psi}(s,a)^2} + \log(\sqrt{2\pi}\sigma_{\psi}(s,a))\right]
\end{aligned}
\tag{15}
$$

where we used the definition $y_Z = r + \gamma(Z_{\bar{\psi}}(s',a') - \alpha\log\pi_{\bar{\phi}}(a'|s'))$.

Now, let's assume the prior for parameters $\psi$ is a Gaussian distribution with mean zero and variance $\sigma^2$. Then, by Bayes rule, we have

$$
\begin{aligned}
-\log p(\psi\,|\,B) &= -\log p(B\,|\,\psi) - \log p(\psi) + \log p(B) \\
&= \frac{1}{2}\mathbb{E}_{(s,a,r,s')\sim B}\big[(y_Z - Q_{\psi}(s,a))^2\big] + \frac{\lambda}{2}\|\psi\|^2 + C,
\end{aligned}
\tag{16}
$$

where $C$ is constant and $\lambda = 1/\sigma^2$. For simplicity of the analysis, let us assume that the variance $\sigma_{\psi}^2$ is a constant. Then, Equation 15 can be written as

$$L_{\mathcal{Z}}(\psi) = \mathbb{E}_{\substack{(s,a,r,s')\sim B, a'\sim\pi_{\bar{\phi}}, \\ Z(s',a')\sim\mathcal{Z}_{\bar{\psi}}(\cdot\,|\,s',a')}} \left[\frac{(y_Z-Q_{\psi}(s,a))^2}{C_1} + C_2\right], \tag{17}$$

where $C_1$ and $C_2$ are constants.

Combining Equation 16 and Equation 17, we have that $L_{\mathcal{Z}}(\psi) \propto -\log p(\psi\,|\,B)$ and thus consequently we have:

$$p(\psi\,|\,B) = \frac{1}{Z}\exp(-L_{\mathcal{Z}}(\psi)), \tag{18}$$

where $Z$ is the normalizing constant.

**Proposition B.1.** *The objective function in Equation 13 for learning distributional critic is equivalent to the following:*

$$L_{\mathcal{Z}}(\psi) = -\mathbb{E}_{\substack{(s,a,r,s')\sim B, a'\sim\pi_{\bar{\phi}}, \\ Z(s',a')\sim\mathcal{Z}_{\bar{\psi}}(\cdot\,|\,s',a')}} \big[\log\mathbb{P}(\mathcal{T}^{\pi_{\bar{\phi}}}Z(s,a)\,|\,\mathcal{Z}_{\psi}(\cdot\,|\,s,a))\big]$$

*Proof.* The proof is adapted from Ma et al. (2020). From Equation 13, the loss function for distributional critic update is given by

$$L_{\mathcal{Z}}(\psi) = \mathbb{E}_{(s,a)\sim B} D_{\mathrm{KL}}(\mathcal{T}^{\pi_{\bar{\phi}}} \mathcal{Z}_{\bar{\psi}}(s,a) \| \mathcal{Z}_{\psi}(s,a))$$

$$= \mathbb{E}_{(s,a)\sim B}\left[ \sum_{\mathcal{T}^{\pi_{\bar{\phi}}} Z(s,a)} \mathbb{P}(\mathcal{T}^{\pi_{\bar{\phi}}} Z(s,a) \,|\, \mathcal{T}^{\pi_{\bar{\phi}}} \mathcal{Z}_{\bar{\psi}}(\cdot \,|\, s,a)) \log \frac{\mathbb{P}(\mathcal{T}^{\pi_{\bar{\phi}}} Z(s,a) \,|\, \mathcal{T}^{\pi_{\bar{\phi}}} \mathcal{Z}_{\bar{\psi}}(\cdot \,|\, s,a))}{\mathbb{P}(\mathcal{T}^{\pi_{\bar{\phi}}} Z(s,a) \,|\, \mathcal{Z}_{\psi}(\cdot \,|\, s,a))} \right]$$

$$= -\mathbb{E}_{(s,a)\sim B}\left[ \sum_{\mathcal{T}^{\pi_{\bar{\phi}}} Z(s,a)} \mathbb{P}(\mathcal{T}^{\pi_{\bar{\phi}}} Z(s,a) \,|\, \mathcal{T}^{\pi_{\bar{\phi}}} \mathcal{Z}_{\bar{\psi}}(\cdot \,|\, s,a)) \log \mathbb{P}(\mathcal{T}^{\pi_{\bar{\phi}}} Z(s,a) \,|\, \mathcal{Z}_{\psi}(\cdot \,|\, s,a)) \right] + c$$

$$= -\mathbb{E}_{(s,a)\sim B}\left[ \mathbb{E}_{T^{\pi_{\bar{\phi}}} Z(s,a) \sim \mathcal{T}^{\pi_{\bar{\phi}}} \mathcal{Z}_{\bar{\psi}}(\cdot \,|\, s,a)} \log \mathbb{P}(\mathcal{T}^{\pi_{\bar{\phi}}} Z(s,a) \,|\, \mathcal{Z}_{\psi}(\cdot \,|\, s,a)) \right] + c$$

$$= -\mathbb{E}_{(s,a)\sim B}\left[ \mathbb{E}_{\substack{(r,s')\sim B, a'\sim \pi_{\bar{\phi}}, \\ Z(s',a')\sim \mathcal{Z}_{\bar{\psi}}(\cdot \,|\, s',a')}} \log \mathbb{P}(\mathcal{T}^{\pi_{\bar{\phi}}} Z(s,a) \,|\, \mathcal{Z}_{\psi}(\cdot \,|\, s,a)) \right] + c$$

$$= -\mathbb{E}_{\substack{(s,a,r,s')\sim B, a'\sim \pi_{\bar{\phi}}, \\ Z(s',a')\sim \mathcal{Z}_{\bar{\psi}}(\cdot \,|\, s',a')}}\left[ \log \mathbb{P}(\mathcal{T}^{\pi_{\bar{\phi}}} Z(s,a) \,|\, \mathcal{Z}_{\psi}(\cdot \,|\, s,a)) \right] + c$$

where $c$ is a term independent of $\psi$. This completes the proof.

$\square$

## C  DMC EXPERIMENT RESULTS

For DMC (Tassa et al., 2018), we consider 12 hard exploration tasks with both dense and sparse rewards. We refer the readers to Table 3 for the list of these 12 tasks and their corresponding properties. From Table 2 and Figure 8, we see that LSAC outperforms both model-free (DSAC-T (Duan et al., 2023), DIPO (Yang et al., 2023), TD3 (Fujimoto et al., 2018), PPO (Schulman et al., 2017), SAC (Haarnoja et al., 2018a), TRPO (Schulman et al., 2015), DrQ-v2 (Yarats et al., 2022)) and model-based (Dreamer (Hafner et al., 2020)) in 9 out of 12 tasks.

| Environments\Methods | LSAC (ours) | DSAC-T | DIPO | SAC | TD3 | PPO | TRPO | Dreamer | DrQ-v2 |
|---|---|---|---|---|---|---|---|---|---|
| Cheetah Run | **967 ± 98** | 540 ± 173 | 521 ± 112 | 693 ± 191 | 549 ± 138 | 492 ± 76 | 603 ± 48 | 792 ± 168 | 747 ± 172 |
| Cartpole Bal. Sp. | **1000 ± 11** | **1000 ± 24** | 1000 ± 37 | 100 ± 8 | 991 ± 12 | 997 ± 6 | 996 ± 9 | 1000 ± 5 | 1000 ± 11 |
| Cartpole Swi. Sp. | 610 ± 41 | 573 ± 102 | 112 ± 24 | 78 ± 23 | 65 ± 32 | 164 ± 18 | 42 ± 15 | 751 ± 142 | **783 ± 52** |
| Reacher Easy | **941 ± 23** | 928 ± 46 | 931 ± 39 | 912 ± 41 | 905 ± 20 | 249 ± 11 | 331 ± 30 | 481 ± 42 | 938 ± 34 |
| Reacher Hard | **981 ± 142** | 11 ± 4 | 501 ± 182 | 243 ± 91 | 38 ± 14 | 9 ± 3 | 8 ± 1 | 852 ± 152 | 861 ± 109 |
| Fish Swim | **892 ± 122** | 716 ± 59 | 816 ± 65 | 531 ± 92 | 748 ± 101 | 117 ± 12 | 283 ± 44 | 509 ± 38 | 870 ± 74 |
| Hopper Hop | **557 ± 71** | 344 ± 89 | 27 ± 2 | 10 ± 4 | 13 ± 8 | 7 ± 3 | 10 ± 2 | 407 ± 31 | 392 ± 113 |
| Finger Turn Easy | **978 ± 64** | 751 ± 139 | 548 ± 107 | 388 ± 41 | 525 ± 73 | 373 ± 62 | 379 ± 28 | 965 ± 83 | 972 ± 79 |
| Finger Turn Hard | **949 ± 34** | 743 ± 42 | 271 ± 161 | 251 ± 27 | 252 ± 21 | 250 ± 11 | 253 ± 34 | 790 ± 21 | 832 ± 79 |
| Walker Walk | 964 ± 20 | 913 ± 28 | 929 ± 37 | **968 ± 47** | 917 ± 12 | 532 ± 9 | 261 ± 187 | 752 ± 41 | 965 ± 34 |
| Walker Run | **869 ± 67** | 731 ± 93 | 391 ± 227 | 465 ± 71 | 388 ± 86 | 89 ± 21 | 221 ± 33 | 726 ± 85 | 717 ± 39 |
| Quadruped Run | 473 ± 41 | 16 ± 3 | 157 ± 94 | 72 ± 25 | 21 ± 19 | 113 ± 27 | 11 ± 5 | 788 ± 139 | **753 ± 58** |

Table 2: Maximum average return of LSAC and baselines across 10 seeds over 3e6 training steps on selected DMC (Tassa et al., 2018) tasks, which consist of complex control tasks that feature complex dynamics, sparse rewards, and hard explorations. Maximum value and corresponding 90% confidence interval for each task is shown in bold.

## D  GOAL-REACHING MAZE EXPERIMENTS

In this section, we describe the `PointMaze_Medium-v3` and the `AntMaze-v4` environments in detail.

**`PointMaze_Medium-v3` Environment.** In `PointMaze_Medium-v3`, the agent is tasked with manipulating a ball to reach some unknown goal position in the maze, with each observation as a dictionary, consisting of an array of the ball's kinetic information, an `achieved_goal` key representing the current state of the green ball, and a `desired_goal` key representing the final goal to be achieved. The action is a force vector applied to the ball. The initial state of the ball is at the center of the maze, and we define the goal positions to be at the upper-right and lower-left corners. The reward function is defined to be the negative Euclidean distance between the desired goal and the visited state.

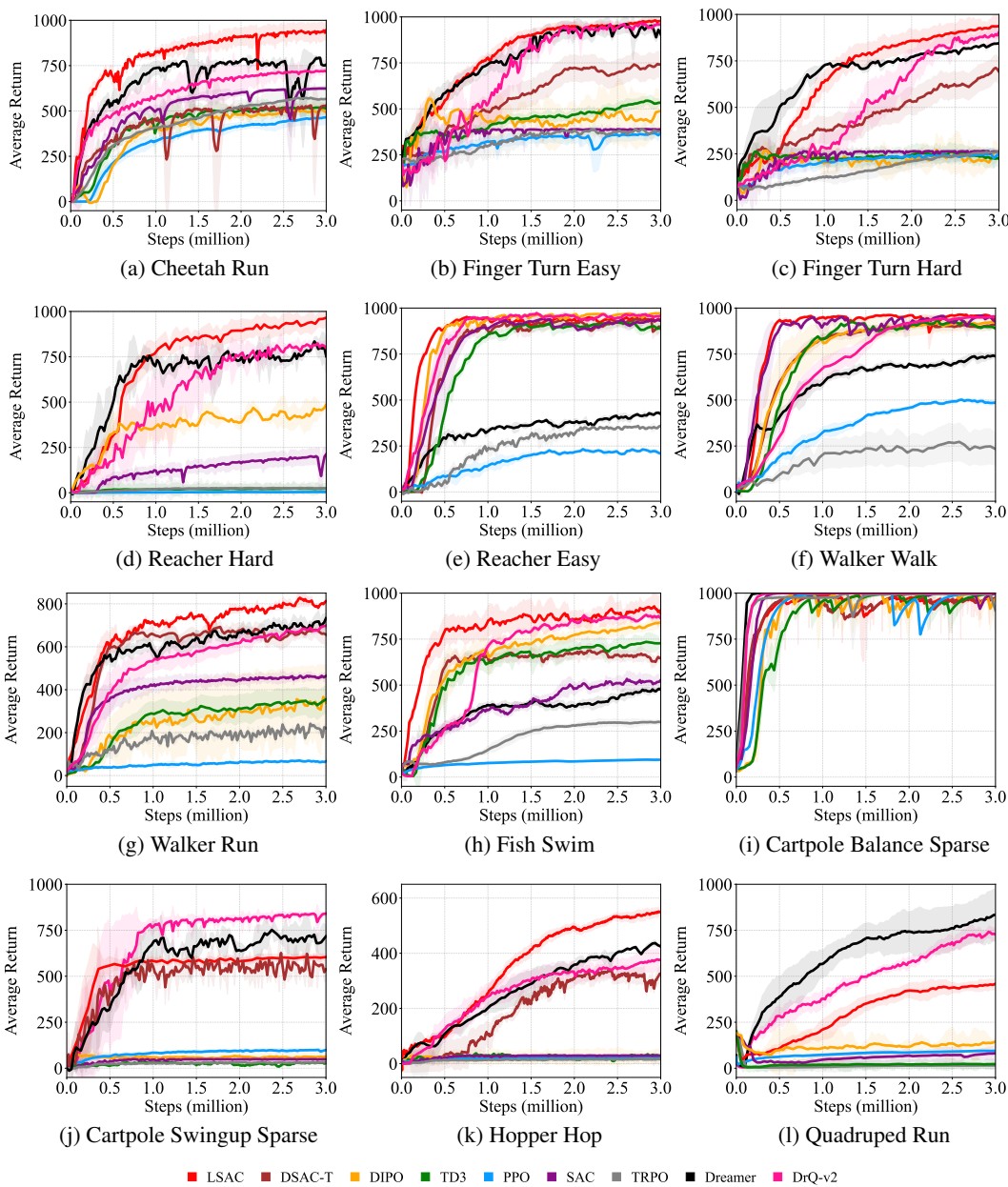

Figure 8: Training curves for 12 DMC (Tassa et al., 2018) continuous control tasks. Results are gathered throughout 3e6 steps of online interaction with the environments, averaged over a window size of 20 and across 10 seeds. Solid lines represent the median performance, and the shaded regions correspond to 90% confidence interval.

**AntMaze-v4 Environment.**    Similar to PointMaze_Medium-v3, AntMaze-v4 is also a navigation task, in which the agent is tasked with controlling a complex 8 degree-of-freedom (DOF) quadruped robot. The objective is to reach one of the two goal positions where the red balls are located. Each goal can be accessed through two routes. The agent can bypass the obstacle on the right by either going up or down, and similarly for the two obstacles on the left. The episode length is set to 700. A sparse 0-1 reward is applied upon reaching the goal.

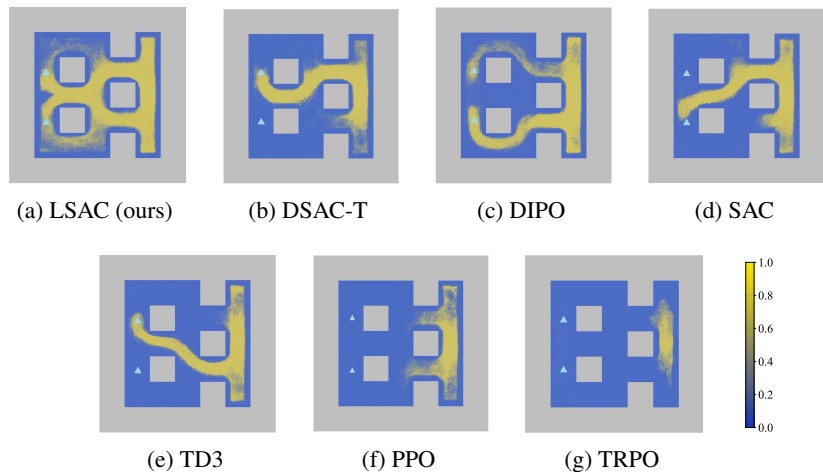

Figure 9: Exploration density maps of LSAC and baseline algorithms tested on the `AntMaze-v4` environment. The two goals are depicted by the triangle markers which are located on the left side of the maze. The starting position is at the center of the right side of the maze.

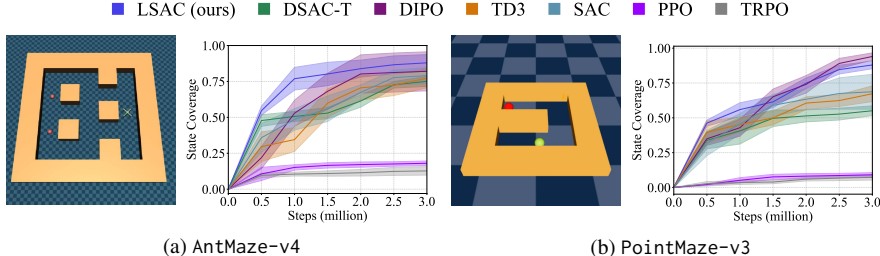

Figure 10: State coverage of LSAC and baselines in the maze environments. For state coverage, we measure the binary coverage of each cell. The y-axis indicates the percentage of cells that have been visited.

# E  IMPLEMENTATION DETAILS FOR THE EXPERIMENTS

In this section, we provide further details on the environments and the algorithm implementation.

## E.1  ENVIRONMENTS

**MuJoCo and DMC.**  Our experimental evaluations are mainly based on the MuJoCo environments (Brockman et al., 2016) and the DeepMind Control Suite (DMC) (Tassa et al., 2018). For MuJoCo, we pick the six most challenging vector-input-based control tasks (HalfCheetah, Ant, Humanoid, Walker2d, Hopper, and Swimmer). On the other hand, DMC offers environments of various difficulty levels, including complex multi-joint bodies and high degree of freedom settings. Hence, we select in particular sparse and hard environments to test the exploration ability of LSAC and baselines. No modifications were made to the state, action, or reward spaces. The action space $\mathcal{A}$ in both continuous control benchmarks is by default the box $[-a_l, a_h]^d$, where $d := \dim \mathcal{A}$ is the dimension of the action space, and $a_l, a_h$ are the low and high action limit, respectively. DMC environment observations, different from MuJoCo's vectorized states input, are stacks of three consecutive RGB images, each of size $84 \times 84$, stacked along the channel dimension to enable inference of dynamic information like velocity and acceleration. Thus, DMC places more emphasis than MuJoCo on pixel-based proprioceptive tasks, which better tests an agent's learning in visual continuous control together with more challenging exploration tasks.

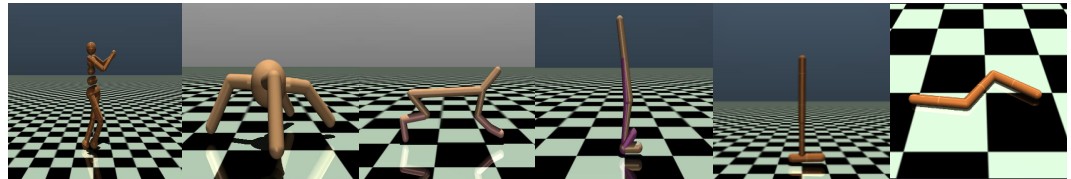

Figure 11: OpenAI MuJoCo (Brockman et al., 2016) benchmarks. From left to right: Humanoid, Ant, HalfCheetah, Walker2d, Hopper, and Swimmer.

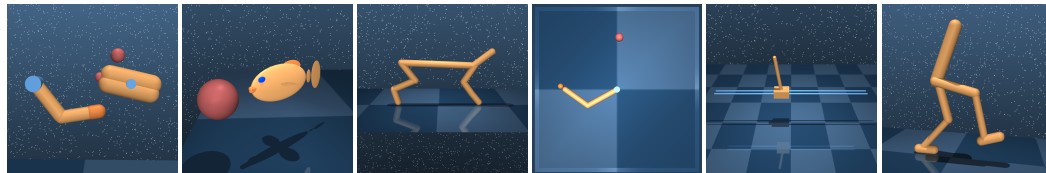

Figure 12: DeepMind Control Suite (Tassa et al., 2018) domains. From left to right: Finger, Fish, Cheetah, Reacher, Cartpole, and Walker.

For DMC, we consider 12 hard exploration tasks with both dense and sparse rewards, in which many other off-policy model-free RL algorithms often struggle. We refer the reader to Table 3 for detailed description of the DMC environments we use in our experiments. Figure 12 shows some examples of the DMC environments.

| Task | Traits | $\dim(\mathcal{S})$ | $\dim(\mathcal{A})$ |
|---|---|---|---|
| Cheetah Run | run, dense | 18 | 6 |
| Finger Turn Easy | turn, sparse | 6 | 2 |
| Finger Turn Hard | turn, sparse | 6 | 2 |
| Reacher Hard | reach, dense | 4 | 2 |
| Reacher Easy | reach, dense | 4 | 2 |
| Walker Walk | walk, dense | 18 | 6 |
| Walker Run | run, dense | 18 | 6 |
| Fish Swim | swim, dense | 26 | 5 |
| Cartpole Balance Sparse | balance, sparse | 4 | 1 |
| Cartpole Swingup Sparse | swing, sparse | 4 | 1 |
| Hopper Hop | move, dense | 14 | 4 |
| Quadruped Run | run, dense | 56 | 12 |

Table 3: A detailed description of each task used in the DeepMind Control Suite (DMC) (Tassa et al., 2018) experiment.

### E.2 BASELINE METHODS

For the evaluation of the baseline algorithms on MuJoCo, we consider DSAC-T (Duan et al., 2023), DIPO (Yang et al., 2023), SAC (Haarnoja et al., 2018a), TD3 (Fujimoto et al., 2018), PPO (Schulman et al., 2017), TRPO (Schulman et al., 2015) and REDQ (Chen et al., 2021). Additionally, for the DMC benchmark we add two additional strong baselines for visual learning, DrQ-v2 (Yarats et al., 2022) and Dreamer (Hafner et al., 2020). In particular, Dreamer is a leading model-based algorithm for visual continuous control setting, which tends to achieve better sample complexity in the expense of a greater computational burden from learning a separate state dynamics model. The highest reported performance on the continuous tasks is achieved by DSAC-T (Duan et al., 2023), an improved version of DSAC (Duan et al., 2021) that uses expected value substituting, twin value distribution learning, and variance-based critic gradient adjusting. Note that since vanilla SAC is a rather weak baseline, we consider augmenting it with the convolutional encoder from SAC-AE (Yarats et al., 2020), similar

to what has been done in (Yarats et al., 2022). We use publicly available codebases for reproducing the baseline results.[1]

The hyperparameters shared across all methods for the MuJoCo tasks are listed in Table 4. We adopted the best hyperparameters reported by the authors of the respective baseline methods, as all of them utilize MuJoCo for continuous control evaluation and conducted extensive hyperparameter sweeps. For implementation simplicity, we maintain these key hyperparameters fixed when testing on the DMC benchmark.

| Hyperparameter | LSAC (ours) | DSAC-T | DIPO | SAC | TD3 | PPO |
|---|---|---|---|---|---|---|
| Num. hidden layers | 3 | 3 | 3 | 3 | 3 | 3 |
| Num. hidden nodes | 256 | 256 | 256 | 256 | 256 | 256 |
| Activation | GeLU | GeLU | Mish | ReLU | ReLU | Tanh |
| Batch size $|B|$ | 256 | 256 | 256 | 256 | 256 | 256 |
| Replay buffer size $|\mathcal{D}|$ | 1e6 | 1e6 | 1e6 | 1e6 | 1e6 | 1e6 |
| Diffusion buffer size $|\mathcal{D}'|$ | 1e6 | N/A | 1e6 | N/A | N/A | N/A |
| Discount for reward $\gamma$ | 0.99 | 0.99 | 0.99 | 0.99 | 0.99 | 0.99 |
| Target smoothing factor $\tau$ | 0.005 | 0.005 | 0.005 | 0.005 | 0.005 | 0.005 |
| Optimizer | aSGLD | Adam | Adam | Adam | Adam | Adam |
| Adaptive bias $(\alpha_1, \alpha_2)$ | $(0.9, 0.999)$ | $(0.9, 0.999)$ | $(0.9, 0.99)$ | $(0.9, 0.99)$ | $(0.9, 0.99)$ | $(0.9, 0.99)$ |
| Inverse temperature $\beta_Q$ | 1e-8 | N/A | N/A | N/A | N/A | N/A |
| Actor learning rate $\beta_\pi$ | 3e-4 | 3e-4 | 3e-4 | 3e-4 | 3e-4 | 7e-4 |
| Number of critics $n$ | 10 | 1 | 1 | 1 | 1 | N/A |
| Critic learning rate $\eta_Q$ | 1e-3 $\rightarrow$ 1e-4 | 3e-4 | 3e-4 | 3e-4 | 3e-4 | 7e-4 |
| Actor Critic grad norm | 0.7 | N/A | 2 | N/A | N/A | 0.5 |
| Replay memory size | 1e6 | 1e6 | 1e6 | 1e6 | 1e6 | 1e6 |
| Entropy coefficient $\alpha$ | 0.2 | 0.2 | N/A | 0.2 | N/A | 0.01 |
| Expected entropy $\overline{\mathcal{H}}$ | $-\dim \mathcal{A}$ | $-\dim \mathcal{A}$ | N/A | N/A | N/A | N/A |
| Diffusion training frequency | 1e4 | N/A | 1 | N/A | N/A | N/A |

Table 4: Common hyperparameters used across all 6 MuJoCo and 12 DMC tasks for LSAC and baselines.

### E.2.1 ADDITIONAL BASELINES FOR THE DMC EXPERIMENT

The two additional baselines we use for the DMC experiments are Dreamer (Hafner et al., 2020) and DrQ-v2 (Yarats et al., 2022). They use a few different network configurations and additional model components. Dreamer (Hafner et al., 2020) uses a pair of convolutional encoder and decoder networks, with remaining functions implemented as three dense layers of size 300 with ELU activations (Clevert, 2015). Action outputs are passed through a `tanh` mean layer, scaled by a factor of 5 and applied with softplus transformation. The world model, value model, and action models are all trained on batches of 50 sequences of length 50, using the Adam (Kingma & Ba, 2014) optimizer with learning rates `6e-4`, `8e-5`, and `8e-5`, respectively. Gradient norms over 100 are scaled down. We note that Dreamer uses an imagination horizon of $H = 15$, which is exclusive to itself, while the value targets $V_\lambda$ are updated with discount factor $\gamma = 0.99$ and $\lambda = 0.95$. The first five episodes are used as warm-up period, where the random actions are sampled with $\mathcal{N}(0, 0.3)$ exploration noise. The latent dynamics model is trained on an information bottleneck objective (Tishby et al., 2000): $\max I(s_{1:T}; (o_{1:T}, r_{1:T}) \,|\, a_{1:T}) - \beta I(s_{1:T}, i_{1:T} \,|\, a_{1:T})$, where $\beta$ is a scalar temperature and $i_t$ are dataset indices such that $p(o_t \,|\, i_t) = \delta(o_t - \bar{o}_t)$. On the other hand, DrQ-v2 (Yarats et al., 2022) uses DDPG (Lillicrap et al., 2015) as a backbone and augment it with $n$-step returns for estimating the TD error. Image encoders $f_\xi$ are used to embed augmented image observations into a low-dimensional latent vector by a CNN. Exploration noise is scheduled according to $\sigma(t) = \sigma_{\text{init}} + (1 - \min(t/T, 1))(\sigma_{\text{final}} - \sigma_{\text{init}})$ at different states of training. A bigger batch size of 512

---

[1]DSAC-T: https://github.com/Jingliang-Duan/DSAC-v2; DIPO: https://github.com/Bellman TimeHut/DIPO; SAC: https://github.com/haarnoja/sac; TD3: https://github.com/sfujim/TD3; PPO: https://github.com/nikhilbarhate99/PPO-PyTorch; TRPO: https://github.com/ikost rikov/pytorch-trpo; REDQ: https://github.com/thu-ml/tianshou; QSM: https://github.c om/Alescontrela/score_matching_rl; Dreamer: https://github.com/danijar/dreamer; DrQ-V2: https://github.com/facebookresearch/drqv2.

and a smaller learning rate of `1e-4` is used. Evaluation frequency is set to be once every 10 episodes, same for each method.

### E.3    MORE ON DIFFUSION $Q$ ACTION GRADIENT

The diffusion generator $\mathcal{M}$ in Algorithm 1 plays a crucial role in synthesizing a batch $B_{M_i}$ of full environment transitions for each critic $Q_{\psi_i}$. These synthetic transitions consist of concatenated states $s_M$, actions $a_M$, episodic rewards $r_M$, next observations $s'_M$, and binary terminal masks $d_M$. We set the mixing ratio of $B_{D_i} \cup B_{M_i}$ to be 0.5 following the design choice of Ball et al. (2023), where $|B_{D_i}| = |B_{M_i}| = 128$. Thus, the overall batch size stays the same as baseline methods at $|B| = 256$. We train $\mathcal{M}$ on normalized transition tuples from the collected online trajectories within the replay buffer $\mathcal{D}$, ensuring that each entry has a mean of zero and a standard deviation of one, except for the "done" signals which remain unnormalized and are thresholded to either 0 or 1 based on a cutoff of 0.5.

To speed up online RL training, we update $\mathcal{M}$ using data from $\mathcal{D}$ every `1e4` online interaction steps and generate `1e6` transitions right after each update of $\mathcal{M}$, which are then stored in the diffusion buffer. The diffusion buffer $\mathcal{D}'$ differs from the online replay buffer $\mathcal{D}$ in that the log probability of Max-Ent actions are not collected and that its action samples are normalized, such that $a_M \in [-1, 1]^d$ to suit for more effective $Q$ action gradient regularization. Our implementation of $\mathcal{M}$ uses the default training hyperparameters in Lu et al. (2024).

Although Lu et al. (2024) has observed the fidelity of these synthetic samples by comparing their high-level statistics to those of the on-policy data, a potential limitation arises from the generated data becoming stale during the `1e4` online interaction steps, due to the fact that $\mathcal{M}$ remains unchanged and the uncertainty in critic function updates. Hence, we apply $Q$ action gradient on normalized synthetic action samples by gradient ascent along the $Q$ gradient field. We make use of the Adam optimizer (Kingma & Ba, 2014) with Polyak coefficients $(\alpha_1, \alpha_2) = (0.9, 0.99)$ and a learning rate of `3e-4`. After $Q$ action gradient update on $a_M \sim \mathcal{D}'$, we replace these state-action pairs in the diffusion buffer to update our belief about the current high-value regions in the action space and mitigate the risk of data staleness.

In Figure 14, we show distribution heatmaps of sampled actions $a$ in the online replay buffer $\mathcal{D}$, synthesized action samples $a_M$ in the diffusion buffer $\mathcal{D}'$, as well as $Q$ gradient-optimized actions $a_M \sim \nabla_a Q$. We observe that one update of $\mathcal{M}$ every `1e4` steps is adequate to match the high-valued region in the sample distributions. We also plot the best average return corresponding to each choice of updating frequency in three MuJoCo environments in Figure 13, which supports our choice of update frequency of $\mathcal{M}$.

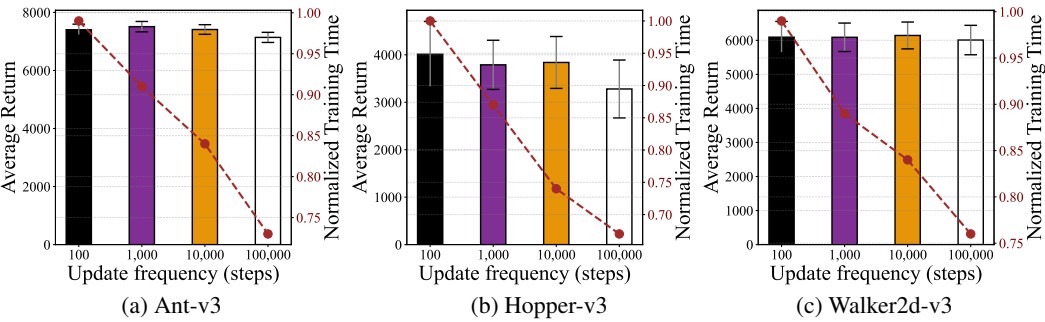

|  |  |  |
|:--:|:--:|:--:|
| (a) Ant-v3 | (b) Hopper-v3 | (c) Walker2d-v3 |

Figure 13: Sensitivity analysis on the performance and training time of LSAC for different choices of update frequencies of diffusion generator $\mathcal{M}$, in three MuJoCo environments. Results are averaged over 10 seeds. The performance difference between each update frequency of $\mathcal{M}$ is not significant on Ant-v3 and Walker2d-v3. However, LSAC scores less with the least update frequency (one in `1e5` steps) in Hopper-v3. The purple dots represent normalized training time.

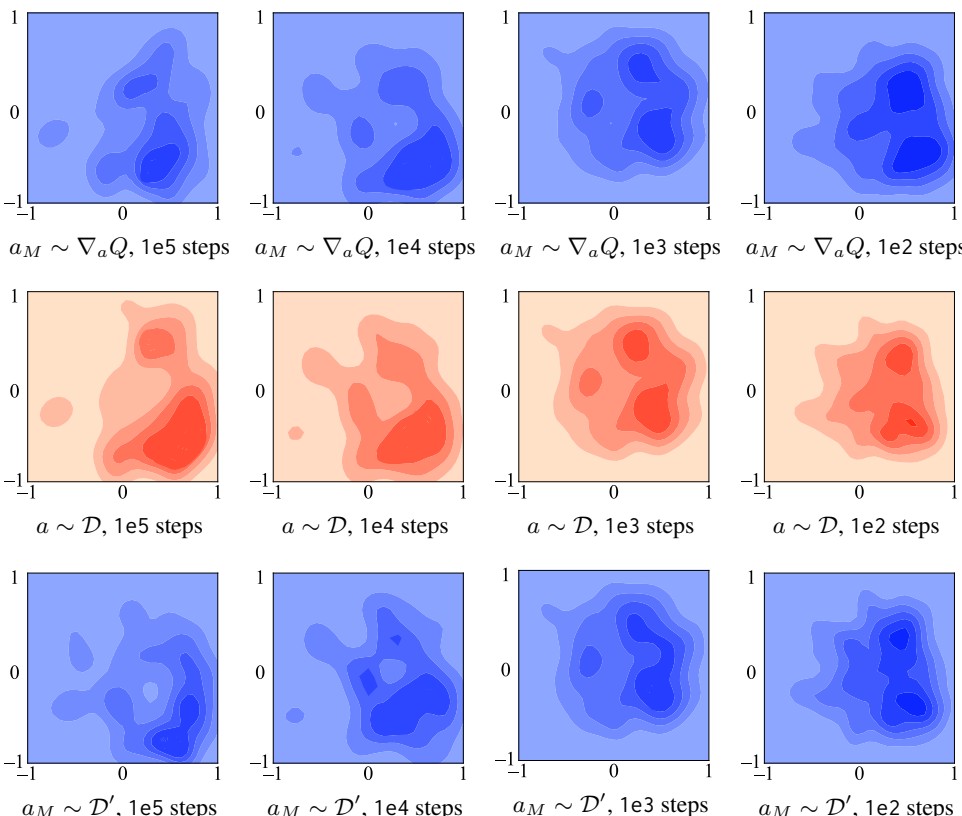

Figure 14: Comparison of $Q$ gradient optimized sample distribution $d_{\{a_M \sim \nabla_a Q\}}$ (first row), replay action sample distribution $d_{\{a \sim \mathcal{D}\}}$ (second row), and diffusion buffer sample distribution $d_{\{a_M \sim \mathcal{D}'\}}$ (third row). The number of steps in the caption means the update frequency of $\mathcal{M}$. Darker regions denote higher density. We randomly sample a batch of 1e3 transitions from all three distributions at 2e5 time steps on the Ant-v3 environment. The 2D kernel density estimate of these distributions are taken from action space dimensions 2 and 3. We see that updating $\mathcal{M}$ every 1e4 steps is suitable for replicating the on-policy buffer action distributions with an optimal total training time. More frequent updates of $\mathcal{M}$ during training (frequency $\leq$ 1e3 steps) leads to higher training time cost as shown in Figure 13. $Q$ gradient optimization further refines the synthesized actions $a_M \sim \mathcal{D}'$ to better align with $d_{\{a \sim \mathcal{D}\}}$.

## F    COMPUTATIONAL EFFICIENCY OF LSAC COMPARED TO BASELINES

Ensuring computational efficiency is critical if we want to make application of RL in the real world problems practical. As LSAC introduces parallel distributed critic learning with diffusion synthesized state-action samples for diverse critic learning, it naturally raises a question of whether the processing speed (wall-clock runtime) and the memory efficiency (number of learnable parameters in each model) become a bottleneck for training LSAC. To answer this question, we compare LSAC to other baselines representative of two other RL training paradigms namely single-critic learning with policy entropy and diffusion policy inference. Among our baselines, we use DSAC-T (Duan et al., 2023), SAC (Haarnoja et al., 2018a), DIPO (Yang et al., 2023), QSM (Psenka et al., 2024), and REDQ (Chen et al., 2021). Our experiment indicates that, besides demonstrating superior sample efficiency and outperforming the baselines in most environments in Figure 1 and Figure 8, LSAC also achieves comparable or better computational efficiency than that of the baselines, as shown in Figure 15. To facilitate fair wall-clock time comparison, all algorithms are trained on the same hardware (i.e a single NVIDIA Quadro RTX 8000 GPU machine).

From Table 5 and Figure 15, we see that the processing speed of LSAC is somewhat slower than that of DSAC-T, primarily due to the time spent on parallel critic learning and diffusion upsampling, which is amortized over every 1e4 steps. However, LSAC training is significantly faster than ensemble based method – REDQ. As for diffusion-based policy methods like DIPO and QSM, while they offer the benefit of multimodal action distributions, they do so at the cost of expensive diffusion policy inference steps during online trajectory rollouts, which makes them considerably slower during training. A clear comparison of total time taken by these algorithms can be found in Table 5 and Figure 15.

| Environment\Algorithm | LSAC (ours) | DSAC-T | DIPO | SAC | QSM | REDQ |
|---|---|---|---|---|---|---|
| Ant-v3 | 1128 (1095) | 876 (857) | 2119 (911) | 751 (722) | 1898 (735) | 1623 (1589) |
| HalfCheetah-v3 | 1157 (1112) | 858 (799) | 2253 (946) | 781 (737) | 1912 (779) | 1740 (1694) |
| Walker2d-v3 | 1141 (1101) | 901 (879) | 2191 (966) | 761 (721) | 1850 (753) | 1828 (1625) |
| Humanoid-v3 | 1209 (1162) | 904 (857) | 2410 (911) | 768 (719) | 1937 (732) | 1794 (1633) |
| Hopper-v3 | 1124 (1078) | 884 (792) | 2391 (898) | 772 (721) | 1893 (778) | 1879 (1417) |
| Swimmer-v3 | 1197 (1049) | 910 (891) | 2201 (913) | 764 (703) | 1829 (716) | 1803 (1544) |

Table 5: Process times in ms per each actor-critic update loop in the MuJoCo environments. Process times per $Q$ functions update are shown in the parenthesis.

| Environment\Algorithm | LSAC (ours) | DSAC-T | DIPO | SAC | QSM | REDQ |
|---|---|---|---|---|---|---|
| Ant-v3 | 2.992M | 301K | 5.109M | 168K | 4.829M | 1.066M |
| HalfCheetah-v3 | 2.591M | 269K | 4.307M | 177K | 3.516M | 961K |
| Walker2d-v3 | 2.447M | 255K | 4.092M | 146K | 3.978M | 795K |
| Humanoid-v3 | 4.807M | 472K | 7.041M | 166K | 6.125M | 1.840M |
| Hopper-v3 | 2.944M | 243K | 4.121M | 146K | 3.772M | 796K |
| Swimmer-v3 | 2.874M | 239K | 4.908M | 135K | 4.063M | 771K |

Table 6: Number of learnable parameters in each method. LSAC has less parameters compared to diffusion policy-based model-free algorithms (DIPO, QSM) but more than those of double or ensemble-based actor-critic methods (DSAC-T, SAC, REDQ). LSAC uses the same network structure and ensemble size across tested environments. The difference in the number of parameters is due to different observation and action dimensions in each environment, which affect the layer sizes of model networks.

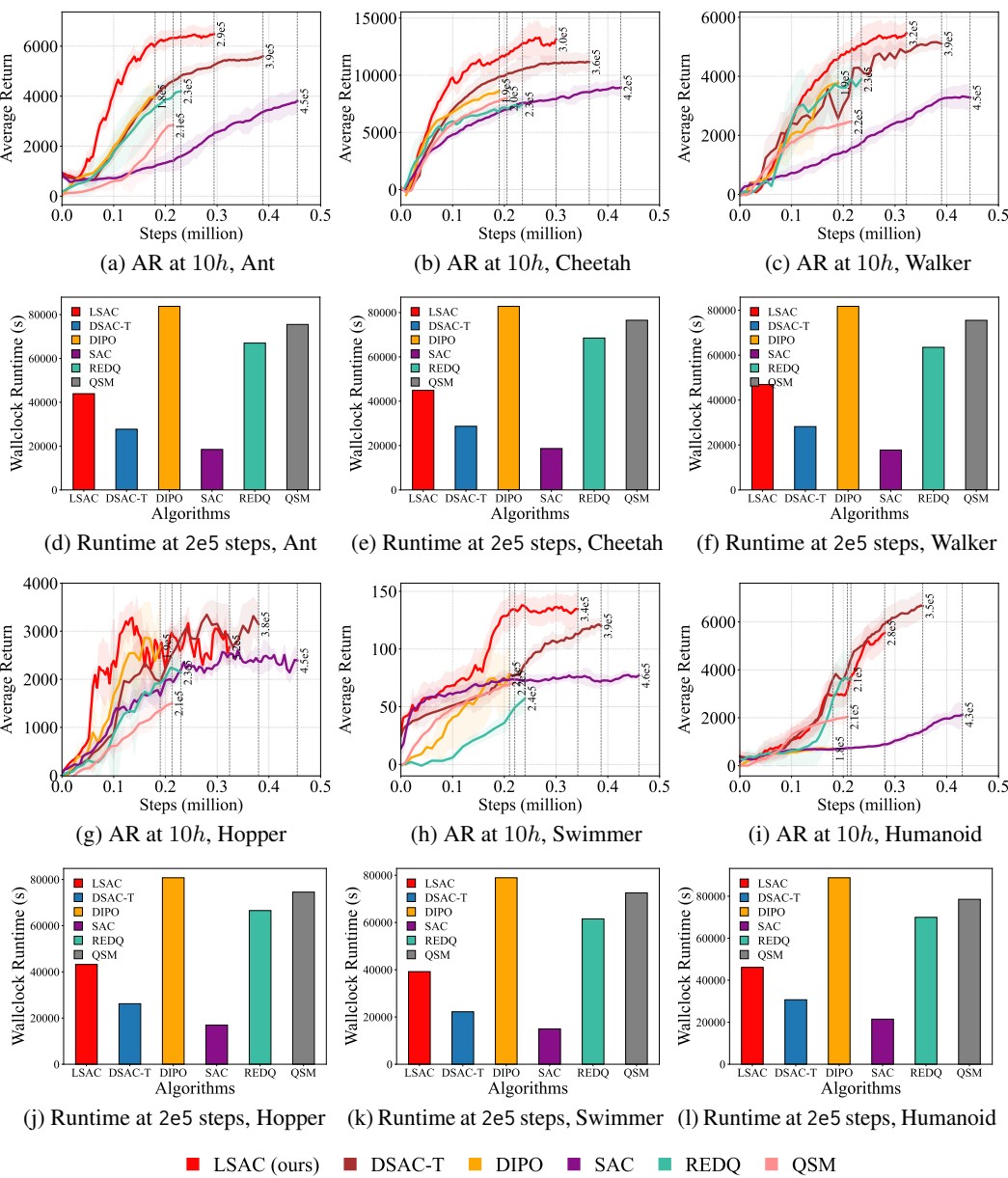

Figure 15: The first and the third row show the number of steps completed in 10 hour of training and the corresponding average return (AR) in corresponding environments. The second and the fourth row show the average wall-clock runtime in seconds to complete 2e5 steps in respective environments.

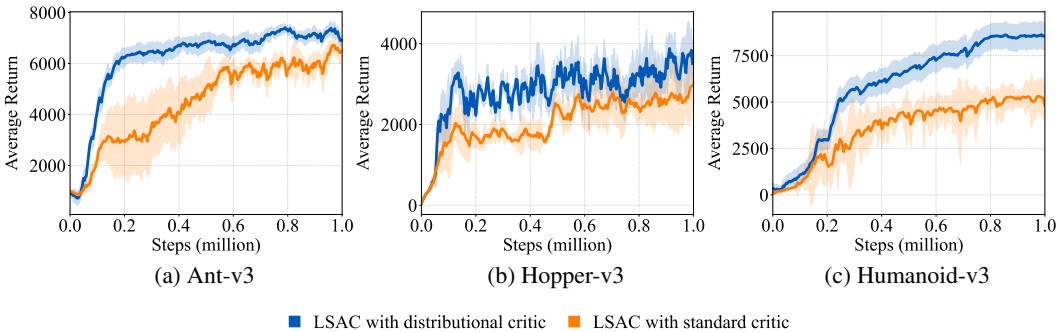

Figure 16: Ablation on MuJoCo environments comparing the replacement of the distributional critic component with a standard critic. LSAC is more performant with improved stability than the ablated counterpart.

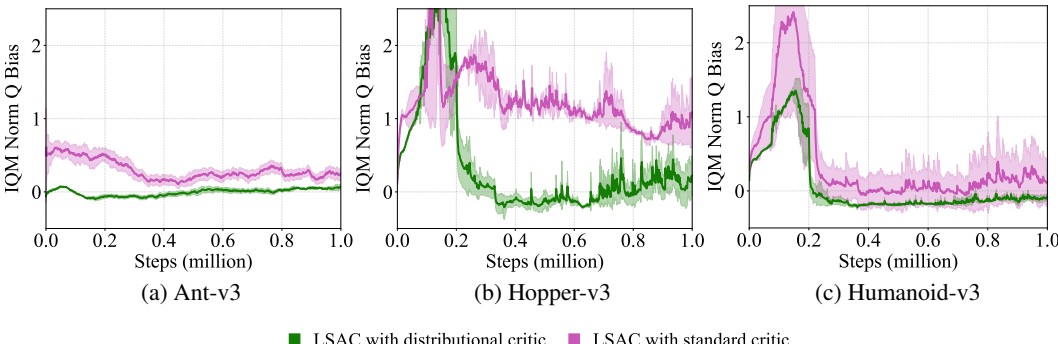

Figure 17: Normalized $Q$ bias plots for ablation study of the distributional critic component in LSAC. The $Q$ bias value is estimated using the Monte-Carlo return over 1e3 episodes on-policy, starting from states sampled in the replay buffer.

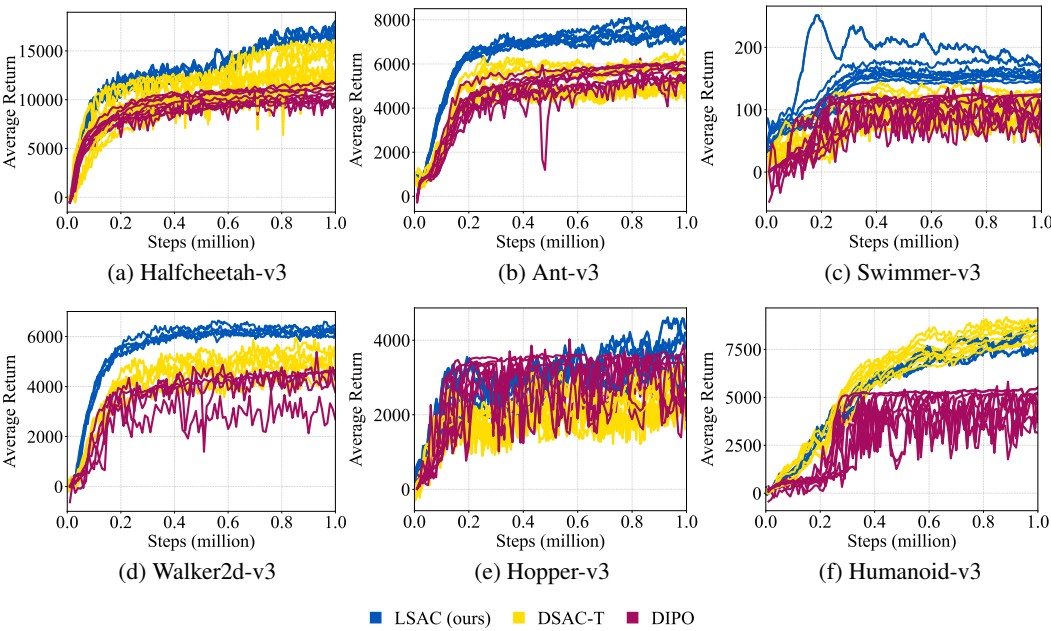

Figure 18: Training curves for the six most challenging MuJoCo continuous control tasks over 1e6 time steps and over 10 individual runs. LSAC is almost similarly stable as DSAC-T, and more stable compared to DIPO in most cases.

