# OpenReview forum: "Langevin Soft Actor-Critic: Efficient Exploration through Uncertainty-Driven Critic Learning"
_ICLR.cc/2025/Conference — ICLR 2025 Poster_

### Official Review · Reviewer_j9KN · 2024-10-25

**Soundness:** 3
**Presentation:** 3
**Contribution:** 3
**Rating:** 8
**Confidence:** 4

**Summary:**

This paper presents a novel model-free RL algorithm, Langevin Soft Actor Critic (LSAC), which enhances critic learning through uncertainty estimation rather than just optimizing the policy. LSAC features three innovations: approximate Thompson sampling via distributional Langevin Q updates, parallel tempering for exploring multiple posterior modes, and diffusion-synthesized state-action samples regularized with Q action gradients. Experiments show that LSAC outperforms or matches mainstream model-free RL algorithms in continuous control tasks and is the first successful application of Langevin Monte Carlo-based Thompson sampling in continuous action spaces.

**Strengths:**

- This paper uses LMC combined with distributed Q to learn critic, and also uses the action gradient method to improve the diversity of samples in the buffer. It is an interesting work;
- The experiments in this paper are very comprehensive, and it has achieved very good performance in many environments, and the curve rises very quickly;
- Compared with diffusion as a policy, this paper does not have the risk of too long sampling time in actual use.

**Weaknesses:**

- In the related work, I strongly suggest you add the work on combining diffusion with online reinforcement learning. Although they focus on the policy, they are also directly related to your paper. Eg. Policy Representation via Diffusion Probability Model for Reinforcement Learning (Yang et al.), Learning a Diffusion Model Policy from Rewards via Q-Score Matching (Psenka et al.), Diffusion Actor-Critic with Entropy Regulator (Wang et al.), etc. Please write a paragraph in the paper to summarize.

- I don't agree with the paragraph around line 68. Methods like SAC and DSAC-T are soft policies. They balance exploration and utilization by introducing policy entropy. How can this be classified as a trick? I think their core is also heuristic, which is essentially the same category as your method.

- In Algorithm 1, there are 4 for loops for each policy update, and the time-consuming action gradient is used, but the time efficiency is not analyzed, and the performance difference brought by different action gradient steps is not analyzed. Please add time efficiency analysis: for example, when training for 8 hours at a time, compare the performance of the algorithms (draw a curve with time as the horizontal axis and TAR as the vertical axis).

- You don’t seem to have experimentally proved some of the problems caused by high-dimensional action space. Compare your current method with the previous method to see why this problem disappears (analyze the experimental results).

- Based on the citations of DSAC-T, I found a work Diffusion Actor-Critic with Entropy Regulator (Wang et al.). Please compare their performance numerically.

- I think the core work in Q-update is to combine Provable and practical: Efficient exploration in reinforcement learning via Langevin Monte Carlo with DSAC-T: Distributional Soft Actor-Critic with Three Refinements. This way of updating Q can indeed alleviate overestimation while bringing better performance, but the novelty may be less.

I'll consider score changes based on your response.

**Questions:**

- In line 9 of Algorithm1, you directly replaced $a_M$ with $a_{M'}$, which destroyed the Markov chain. What problems exist in fusing such data with $B_D$? Please analyze it theoretically.

- Algorithm1 Why doesn't the strategy also choose to update on $B_i$?

- Line341 Is the frequency of updating the diffusion model too low? Does the data generated in this way really have many benefits for sample distribution? Please analyze the sample distribution, as well as the performance and sample distribution results corresponding to different update frequencies.

- Why do other algorithms in Table1 have no variance? It seems that you didn't say how you calculated the data in the table.

- Do you also sample 20 samples like DSAC-T for each interaction with the environment?

- In the high-dimensional action space task Humanoid-v3, why is your performance much lower than DSAC-T?

---

> ### Author Response · Authors · 2024-11-20
>
> We thank the reviewer for their valuable time and effort in providing detailed feedback on our work. We hope our response will fully address all of the reviewer's points.
>
> ---
>
> ### Related work for online RL with diffusion methods.
>
> Thanks a lot for the suggestion. Indeed, works that combine diffusion models with online RL are related to our paper. We added a paragraph with the title **Online reinforcement learning with diffusion** in our related work. Please note that, due to space constraint, we moved our related work section to Appendix A. We would also like to highlight that we had already included DIPO [1] as one of our baselines. In our revised version, we added additional diffusion policy based baseline QSM [2] in the MuJoCo experiment.
>
> ---
>
> ### Policy entropy and exploration.
>
> Thanks for your comment on this. You are right; classifying design components of DSAC-T, SAC and TD3 as tricks is a bit unfair. Our intention was to convey that, while these heuristics inspired approaches work in practice, they lack theoretically principled exploration mechanisms (such as optimism or Thompson sampling) that are critical to ensure directed or deep exploration. In our revised version, Section 4.2, through a newly added experiment on understanding exploration capability of LSAC and other baselines, we highlight that heuristics such as entropy maximization alone are not sufficient for performing deep exploration (Figure 7 and Figure 9).
>
> The revised paragraph around line 68 looks as follows:
>
>
> ```
> Furthermore, traditional continuous control benchmarks such as DSAC-T (Duan et al., 2023), REDQ (Chen et al., 2021), SAC (Haarnoja et al., 2018), and TD3 (Fujimoto et al., 2018), while benefiting from heuristics such as noise perturbed actions sampling and entropy maximization, do not sufficiently emphasize principled and directed exploration in their design principles. On the flip side, LSAC is highly exploratory in nature and effectively increases state-coverage during training by virtue of using theoretically principled LMC based Thompson sampling.
>  ```
> ---
>
> ### We added time efficiency analysis.
>
> Thanks for your suggestion to add time efficiency analysis. We added a comprehensive time efficiency analysis in Appendix F. The suggested plot can be found in Figure 15.
>
> Regarding for loops in each policy update, we actually have two “real” for loops. Let us clarify. In the original submission version, in Algorithm 1, Line 8, we have a for loop over $J$ for action gradient update. However, in all of our experiments, we set $J = 1$ as we mentioned in line 923 - 925 in the original submission:
>
> ```
> Since a large value of action gradient steps J greatly increases training time, we set J = 1 in LSAC so that all experiments can be finished in a reasonable time.
>
> ```
> Since $J = 1$ already gives strong performance, in our revised paper, we removed this for loop involving $J$.
>
> The for loop in Line 7 of algorithm 1, where we go through each sample in $B_{M_{i}}$ is not technically a loop as it is just optimizing over a batch of samples like one optimizes actor/critic networks in policy/value iteration and such batch operation are usually performed in fairly efficient manner in most standard deep learning libraries such as PyTorch. In the revised edition, we removed this for loop to indicate this part can be done as a batch operation.
>
> ---
>
> ### Problems caused by high-dimensional action space.
>
> > *You don’t seem to have experimentally proved some of the problems caused by high-dimensional action space. Compare your current method with the previous method to see why this problem disappears (analyze the experimental results).*
>
> We are not sure we understand the comment here. Indeed, high-dimensional action space makes any task much more challenging. Would the reviewer kindly suggest what kind of experiment they have in mind for experimental validation of this?
>
> ---

---

> > ### Author Response · Authors · 2024-11-20
> > **Continuation of Response (part 2)**
> >
> > ---
> > ### On not comparing with the paper “Diffusion Actor-Critic with Entropy Regulator” from NeurIPS 2024.
> >
> > We thank the reviewer for pointing out this work [3] that proposes the algorithm DACER. In the revised version, we discussed this work in our related work section. We welcome the suggestion to compare our method against DACER in the experiments. However, we could not find any publicly available training logs of DACER and given the expensive nature of training diffusion policy based methods, it is not possible to run experiments using their code for us during the ICLR discussion period. Moreover, we would like to emphasize that according to [ICLR 2025 Reviewer Guide](https://iclr.cc/Conferences/2025/ReviewerGuide), we are not required to compare our work against this paper as it has only recently been accepted to NeurIPS 2024 and the list of accepted papers of NeurIPS 2024 was made public much later the ICLR full paper deadline of October 1. We highlight the relevant part from the ICLR 2025 Reviewer Guide:
> >
> > ```
> > Q: Are authors expected to cite and compare with very recent work? What about non peer-reviewed (e.g., ArXiv) papers? (updated on 7 November 2022)
> >
> > A: We consider papers contemporaneous if they are published within the last four months. That means, since our full paper deadline is October 1, if a paper was published (i.e., at a peer-reviewed venue) on or after July 1, 2024, authors are not required to compare their own work to that paper. Authors are encouraged to cite and discuss all relevant papers, but they may be excused for not knowing about papers not published in peer-reviewed conference proceedings or journals, which includes papers exclusively available on arXiv. Reviewers are encouraged to use their own good judgement and, if in doubt, discuss with their area chair.
> > ```
> >
> > ---
> >
> > ### On novelty.
> >
> > While the core idea of Q update in LSAC is to perform Langevin Monte Carlo (LMC) update from [4]  with DSAC-T, the idea of combining them and resulting benefit is non-trivial and non-obvious. Moreover, LSAC has several other novel design choices that make it significantly different from [1] and DSAC-T. For example, instead of standard LMC update from [1], we use parallel tempering with LMC for exploring different modes of the Q-posterior. As the reviewer acknowledged in the review summary, this work is the first successful application of LMC based Thompson sampling in continuous action space. Thus, we hope that the contribution of this work to the RL literature would be significant.
> >
> > ---

---

> > > ### Author Response · Authors · 2024-11-20
> > > **Continuation of response (part 3)**
> > >
> > > ---
> > >
> > > ### Answer to questions.
> > >
> > >
> > > * > Algorithm1 Why doesn't the strategy also choose to update on $B_i$?
> > >
> > > Thanks for catching it. Sorry, it was a typo. It should have been $B_i$ and not $B_{D_i}$. We corrected it in the revised version.
> > >
> > > * > Line341 Is the frequency of updating the diffusion model too low? Does the data generated in this way really have many benefits for sample distribution? Please analyze the sample distribution, as well as the performance and sample distribution results corresponding to different update frequencies.
> > >
> > > Thanks for this question and the suggestions to analyze these. We ran a sensitivity analysis on the performance of LSAC for different update frequencies for the diffusion model. We present the result in Figure 6 (c ) and Figure 13. We observe that increasing the update frequency doesn’t really impact on the average return much while increasing the training time significantly.
> > >
> > > 	In Figure 14, we show distribution heatmaps of sampled actions from online replay buffer, diffusion synthesized action samples, and action gradient refined synthetic actions. From the figure, we see that when we perform action gradient on synthetic diffusion generated action, the resulting distribution of refined action matches closely with the action distribution of real online replay buffer.
> > >
> > > *  > In line 9 of Algorithm1, you directly replaced $a_M$ with $a_{M′}$, which destroyed the Markov chain. What problems exist in fusing such data with $B_D$? Please analyze it theoretically.
> > >
> > > 	As described in the last question’s answer, from Figure 14, we see that after performing action gradient operation and replacing $a_M$ with $a_{M′}$, the resulting action distribution gets closer to the action distribution of $B_D$. So, while they are not identical, it still enhances the diversity of state-action pairs and increases the UTD ratio for critic update.
> > >
> > > * > Why do other algorithms in Table1 have no variance? It seems that you didn't say how you calculated the data in the table.
> > >
> > > 	We added variance for all algorithms in Table 1 and Table 2 in the revised version.
> > >
> > > * > Do you also sample 20 samples like DSAC-T for each interaction with the environment?
> > >
> > > 	Yes, for each interaction with the environment, we sample 20 interaction samples.
> > >
> > > * > In the high-dimensional action space task Humanoid-v3, why is your performance much lower than DSAC-T?
> > >
> > > 	From Table 1, we see that in Humanoid-v3, the average return for LSAC is 8545 while DSAC-T has average return of 9028. So, the performance difference is not really much. But yet it’s an interesting observation. It could be due to potential over-exploration in LSAC.
> > >
> > >
> > > ---
> > >
> > > Finally, we have outlined the list of updates made in the revised paper based on feedback from all the reviewers in [Overall Response](https://openreview.net/forum?id=FvQsk3la17&noteId=xFgYGltWDv) above. There, we provided a summary of new experiments and analysis that have been added to the revised paper.
> > >
> > >
> > > We hope we have addressed all of your questions/concerns. If you have any further questions, we would be more than happy to answer them and if you don’t, would you kindly consider increasing your score?
> > >
> > > [1] ​​Yang, Long, et al. "Policy representation via diffusion probability model for reinforcement learning." arXiv preprint arXiv:2305.13122 (2023).
> > >
> > > [2] Psenka, Michael, et al. "Learning a Diffusion Model Policy from Rewards via Q-Score Matching." Forty-first International Conference on Machine Learning. (2024)
> > >
> > > [3] Wang, Yinuo, et al. "Diffusion Actor-Critic with Entropy Regulator." arXiv preprint arXiv:2405.15177 (2024).
> > >
> > > [4] Ishfaq, Haque, et al. "Provable and Practical: Efficient Exploration in Reinforcement Learning via Langevin Monte Carlo." The Twelfth International Conference on Learning Representations. (2024)

---

> ### Comment · Reviewer_j9KN · 2024-11-21
> **Re: authors**
>
> Regarding DACER, I have indeed not seen their training logs, so it is okay not to make a comparison (but they uploaded it to arXiv in May 2024). Novelty is still a concern, but with your explanation and sufficient experiments, this concern can be reduced. Try to put related work in the main text, which is one of the academic norms. Although the training time has become longer, it is still within an acceptable range.
>
> 1. Humanoid is a problem of high-dimensional action space, so it is a bit inconsistent with your statement that it did not show the best performance on this task (you said that you can better solve the task of high-dimensional action space).
> 2. You mentioned that the update frequency of the diffusion model has little effect on performance, which is a bit confusing. Because the update frequency is slow, there should not be a lot of data that can actually play a role in the early and middle stages of training.
>
> I am willing to increase it to 6 points, and consider the score changes based on subsequent answers and conversations with other reviewers.

---

> > ### Author Response · Authors · 2024-11-22
> > **answers to further questions**
> >
> > We thank the reviewer for their positive support on our paper and engaging questions.
> >
> > 1. The challenges highlighted in (C1) in the Introduction are valid for any continuous and multi-dimensional action space when it comes to designing Thompson sampling based algorithms for continuous control setting. So, instead of saying “high dimensional” we revise it as “multidimensional”. The revised paragraph looks as follows:
> > ```
> > (C1) Multidimensional continuous action spaces. In continuous control settings, actions are typically continuous and multidimensional tensors. This makes it computationally intractable to naively select the exact greedy actions based on Q posterior approximations, often leading to sub-optimal performance and inefficient exploration.
> >
> > ```
> > We hope this resolves the minor inconsistency in the earlier version.
> >
> > 2. **Diffusion model update frequency**: Thanks for the question! Even though we are updating the diffusion model every `1e4` steps, note that we run MuJoCo experiments for a total of `1e6` steps and DMC experiments for a total of `3e6` steps. So, even in the MuJoCo experiments, throughout the training process, we update the diffusion model 100 times. So, if we consider the total number of steps, the time period when we have less data, is only the first 1% of the time of total training.
> >
> > We hope we have addressed all of your questions/concerns. If you have any further questions, we would be more than happy to answer them. Finally, we are grateful for the reviewer’s positive support on our paper. Given our extensive additional experiment results as outlined in the [Overall Response](https://openreview.net/forum?id=FvQsk3la17&noteId=xFgYGltWDv) above, if the reviewer thinks the work is worthy of a higher score, we would be immensely grateful.

---

> ### Comment · Reviewer_j9KN · 2024-11-22
>
> Overall this work is an interesting attempt and I would love to see improvements in exploration, which is something that the RL community might be interested in. Given that most of the problems are solved, I am willing to update my score.
>
> However, the article seems a bit too complicated. It is recommended to delete some less important formulas before the final version, or put them in the appendix.

---

> > ### Author Response · Authors · 2024-11-23
> > **Thank you!**
> >
> > We thank the reviewer for reading our response and we are glad that our response addressed all the questions and concerns of the reviewer. Additionally, we would like to convey special thanks to the reviewer for suggesting several new experiments and analysis, in particular the time efficiency analysis and the sample distribution analysis of the data generated by the diffusion model. We believe incorporation of them has strengthened our manuscript much more.
> >
> >  Finally, we thank the reviewer again for their consistent support of our paper.

---

### Official Review · Reviewer_iNev · 2024-11-04

**Soundness:** 2
**Presentation:** 3
**Contribution:** 2
**Rating:** 6
**Confidence:** 3

**Summary:**

This paper introduces a new model-free RL algorithm Langevin Soft Actor Critic (LSAC). It uses Langevin Monte Carlo (LMC) for updating the Q function to help exploration and improve performance, the authors pointed out several key challenges in taking this approach and propose solutions that show benefit to performance in the ablations. When compared to a number of other baseline RL algorithms on MuJoCo and DMC benchmarks, LSAC shows stronger overall performance. This is the the first successful application of a Langevin Monte Carlo (LMC) based Thompson sampling in RL continuous control setting.

**Strengths:**

**originality**

- authors study and show how to do Langevin Monte Carlo (LMC) based Thompson sampling in RL continuous control setting properly and achieve strong performance, this is a novel contribution.

**quality**

- overall good quality of presentation and writing.
- fairness of comparison: I appreciate that the authors discuss comparison fairness and provide full details of the hyperparameter setting in the appendix and give source code.

**clarity**
- overall clear

**significance**
- strong results: Figure 1, table 1 show that proposed method has fairly consistent stronger results compared to other baselines.
- results show how to properly use Langevin Monte Carlo (LMC) based Thompson sampling in DMC and MuJoCo.
- ablations: it is good to have ablations to understand the effect of different design choices.

**Weaknesses:**

Some of the arguments made in the paper seems not adequately supported, some more explanations or experiments will help:

- Line 346: the authors argue LSAC has good performance while being simple in implementation. Why is it simple in implementation? It seems to me it has a number of additional components/hyperparameters compared to a baseline such as SAC, and these extra components seem not trivial to implement correctly. The complexity of the method seems a weakness to me.
- Line 460: The authors argue that Figure 5 shows LSAC as more stable. I don't feel convinced. The fluctuation in its performance seems similar to DSAC-T, and even when compared to DIPO, LSAC has some wild changes in Figure 5 (c).
- Is there a comparison on computation efficiency? With the additionl components in the proposed method, how much computation/wall clock time does it take to finish a run compared to other algorithms?
- Although a lot of details and discussion are given to show an effort to compare to alternative algorithms in a fair manner, it is a bit unclear to me how 2 things are made fair: (1) network capacity, the proposed approach seems to have more critics and additional networks (such as in the diffusion part), how does it compare to other baselines in terms of e.g. number of parameters? And (2) UTD ratio, if the proposed approach is doing more updates, is it still fair to compare it to baselines that take limited updates? I understand that some of them might not work well with more updates due to instability, but in that case, one might argue it would be good to compare to algorithms that do benefit from ensemble of networks and higher UTD such as REDQ?

**Questions:**

- It seems to me a main part of the strong performance comes from a better exploration technique. In the baselines you compared to, are some of them focused on exploration techniques? If we replace the exploration in proposed method with some other recent exploration technique, will we obtain a similar strong performance?

---

> ### Author Response · Authors · 2024-11-20
>
> We thank the reviewer for their valuable time and effort in providing detailed feedback on our work. We hope our response will fully address all of the reviewer's points.
>
> ---
>
> ### Complexity of the method
>
> By simple implementation, we intended to mean that, once the different components of LSAC are pieced together, it would seem to have a simple implementation. We updated the paragraph and the revised paragraph around Line 340 looks as follows:
>
>
> ```
> From Figure 1 and Table 1, we see that LSAC outperforms other baselines in 5 out of 6 tasks from MuJoCo. In Humanoid-v3 even though DSAC-T outperforms LSAC, the difference is marginal. For space constraint we report the DMC result in Appendix C. The learning curves for DMC tasks are presented in Figure 8.
>
>  ```
> However, we would like to highlight that in essence, LSAC has 3 main components: (1) LMC based distributional Q sampler for which we follow the implementation of [1] and [2] with almost no change of relevant hyperparameters, (2) Synthetic data generation using diffusion model for which we follow the implementation of SynthER from [3]. (3) Parallel tempering whose implementation is very similar to standard ensemble based approaches. Once these components are considered (and implemented) individually, after piecing them together, we hope it would not look as complex as it might seem at first glance.
>
> ---
>
>
> ### Stability of LSAC compared to others.
>
> We updated our comment on stability as follows in Line 474-475:
>
> ```
> While off-policy deep RL algorithms are often challenging to stabilize, we found that LSAC is fairly stable as shown in Figure 18.
> ```
>
> While in Figure 18 (c ) for Swimmer, LSAC has a single run that fluctuates from the rest of the 9 runs, in other tasks such as Ant, Walker, Hopper and Humanoid, LSAC seems to be more stable than DIPO. We also updated the caption in the figure to better reflect this observation.
>
> ---
>
> ### We added time efficiency analysis and comparison of network capacity.
>
> Thanks for your suggestion to add time efficiency analysis. We added a comprehensive time efficiency analysis in Appendix F. The comprehensive plot can be found in Figure 15 and a table in Table 5. We also added a detailed table comparing the number of learnable parameters for different algorithms in Table 6.
>
>
> ---
>
> ### Adding REDQ as baseline.
>
> > *UTD ratio, if the proposed approach is doing more updates, is it still fair to compare it to baselines that take limited updates? I understand that some of them might not work well with more updates due to instability, but in that case, one might argue it would be good to compare to algorithms that do benefit from ensemble of networks and higher UTD such as REDQ?*
>
> Thanks for the suggestion. We added REDQ as an additional baseline in our MuJoCo experiments. We would like to note that REDQ is very insufficient in that it uses a fixed UTD ratio of $G=20$, and all its Q-networks get updated 20 times per each policy iteration. LSAC does not explicitly loop $G$ times but still boasts a high UTD ratio $G\gg 1$ by exploiting mixed state-action data from both on-policy and diffusion buffers.
>
> ---
>
> ### Question on exploration.
>
> We use several benchmarks that have exploration specific design components. For example, SAC and DSAC-T use an entropy maximization framework which is supposed to help with learning more exploratory policy. We also use DIPO [4] which learns a multimodal diffusion policy and is claimed to help with exploration. Given REDQ is an ensemble based method that is heavily inspired from bootstrapped DQN [5], we can also consider it to be an exploratory algorithm.
>
> It is an interesting question whether using some other exploration method instead of LMC based Thompson sampling can give similarly strong performance. However, it is not clear how to incorporate existing recent exploration approaches with distributional critic learning.
>
> ---
>
> Finally, we have outlined the list of updates made in the revised paper based on feedback from all the reviewers in [Overall Response](https://openreview.net/forum?id=FvQsk3la17&noteId=xFgYGltWDv) above. There, we provided a summary of new experiments and analysis that have been added to the revised paper.
>
> We hope we have addressed all of your questions/concerns. If you have any further questions, we would be more than happy to answer them and if you don’t, would you kindly consider increasing your score?

---

> > ### Author Response · Authors · 2024-11-20
> > **Continuation of response (part 2)**
> >
> > [1] Ishfaq, Haque, et al. "Provable and Practical: Efficient Exploration in Reinforcement Learning via Langevin Monte Carlo." The Twelfth International Conference on Learning Representations. (2024)
> >
> > [2] Duan, Jingliang, et al. "DSAC-T: Distributional soft actor-critic with three refinements." arXiv preprint arXiv:2310.05858 (2023).
> >
> > [3] Lu, Cong, et al. "Synthetic experience replay." Advances in Neural Information Processing Systems 36 (2024).
> >
> > [4] Yang, Long, et al. "Policy representation via diffusion probability model for reinforcement learning." arXiv preprint arXiv:2305.13122 (2023).
> >
> > [5] Osband, Ian, et al. "Deep exploration via bootstrapped DQN." Advances in neural information processing systems 29 (2016).

---

> > > ### Author Response · Authors · 2024-11-23
> > > **Follow up**
> > >
> > > Dear Reviewer iNev,
> > >
> > > Thank you so much for taking the time to review our paper. We sincerely appreciate your detailed feedback and appraisal of our work, and have carefully addressed your comments in our response and the revised manuscript. With only 3 days remaining for the discussion period, we would be grateful if you could acknowledge receipt of our responses and let us know if you have further questions. We are eager to engage in any further discussions if needed.
> > >
> > > If our responses have addressed all of your questions/concerns, would you kindly consider increasing your score?
> > >
> > > Thanks.\
> > > Authors

---

> > > > ### Comment · Reviewer_iNev · 2024-11-27
> > > > **Thank you for the rebuttal**
> > > >
> > > > I thank the authors for their response and appreciate the new results and analysis. To me, the algorithm's complexity remains a weakness. However, I will update my score based on the fairly strong results demonstrated and the interesting empirical findings in the paper.

---

> > > > > ### Author Response · Authors · 2024-11-27
> > > > > **Thank you!**
> > > > >
> > > > > We thank the reviewer for reading our response and we are glad that the reviewer recognizes our strong results and interesting empirical findings. Finally, we thank the reviewer again for their consistent support of our paper.

---

### Official Review · Reviewer_ETnf · 2024-11-04

**Soundness:** 3
**Presentation:** 3
**Contribution:** 3
**Rating:** 6
**Confidence:** 3

**Summary:**

Authors of this paper introduces a model-free online RL algorithm named Langevin Soft Actor-Critic (LSAC). The algorithm incorporates  Langevin Monte Carlo (LMC) updates for critic learning with parallel tempering and action refinement on diffusion-synthesized trajectories. it also utilizes distributional Q objective, allowing diverse sampling from multimodal Q posteriors. Through experiments on MuJoCo benchmarks, the algorithm is shown to outperform/match the performance of SOTA baselines.

**Strengths:**

Novel ideas of combining LMC updates, parallel tempering, and diffusion-synthesized action refinement in the LSAC algorithm.
By efficiently exploring state-action spaces through LMC based Q updates and multimodal Q posteriors with parallel tempering, LSAC has the potential to learn a better policy in complex environments.
LSAC demonstrates competitive or superior performance against several established baselines on standard continuous control benchmarks.
The methodological choices in LSAC are grounded in theoretical analysis presented in the paper.

**Weaknesses:**

The implementation seems very complex even by reading through the pseudo code.
The computational cost of LSAC lacks discussion in the paper.
Additional parameters and hyper-parameters are introduced, including 10 parallel critics, additional diffusion buffer, inverse temperature, etc, and ablation study shows that LSAC is sensitive to some of the parameters, which may limit its scalability to different applications.

**Questions:**

As all the techniques introduced and incorporated into LSAC and complexity of the implementation, any analysis why LSAC is not working better than DSAC in the humanoid environment (which is also regarded as the one of the most complex environments among the benchmarks) ?
What is the impact of each key component (LMC updates, parallel tempering, diffusion models) on the performance of LSAC? Have you tried to remove one of these components and see the affect the overall effectiveness?

---

> ### Author Response · Authors · 2024-11-20
>
> We thank the reviewer for their valuable time and effort in providing detailed feedback on our work. We hope our response will fully address all of the reviewer's points.
>
> ---
>
> ### Complexity of the algorithm and additional hyperparameters.
>
> We would like to highlight that in essence, LSAC has 3 main components: (1) LMC based distributional Q sampler for which we follow the implementation of [1] and [2] with almost no change of relevant hyperparameters, (2) Synthetic data generation using diffusion model for which we follow the implementation of SynthER from [3] with their default hyperparameters. (3) Parallel tempering whose implementation is very similar to standard ensemble based approaches. Once these components are considered (and implemented) individually, after piecing them together, we hope it would not look as complex as it might seem at first glance.
>
> ---
>
> ### We added computational efficiency analysis and comparison of network capacity.
>
> Thanks for your suggestion to add time efficiency analysis. We added a comprehensive time/computational efficiency analysis in Appendix F. The comprehensive plot can be found in Figure 15 and a table in Table 5. We also added a detailed table comparing the number of learnable parameters for different algorithms in Table 6.
>
>
> ---
>
> ### Performance comparison of LSAC and DSAC-T in humanoid.
>
> From Table 1, we see that in Humanoid-v3, the average return for LSAC is 8545 while DSAC-T has average return of 9028. So, the performance difference is not really  much. But yet it’s an interesting observation. It could be due to potential over-exploration in LSAC.
>
> ---
>
> ### Impact of each key component and ablation studies.
>
> In Section 4.1, we showed several ablation studies of key components of LSAC in the original submission and added additional ablation studies in the revised version. On Figure 3 and Figure 16, we show the impact of using distributional critics in LSAC. We compare LSAC with distributional critics and a version of LSAC where the distributional critics are replaced by  standard critics implementation (such as in SAC). Furthermore, in Figure 4 and Figure 17, we show how distributional critic in LSAC helps mitigating overestimation bias. In Figure 6 (a), we show the impact of parallel tempering with different numbers of parallel critics. Note that, in this Figure the case with $|\Psi| = 1$ is equivalent to not having any parallel tempering. In Figure 6 (b), we compare the performance when critic is updated using LMC based aSGLD sampler for Q values vs when the critic is updated using the standard Adam optimizer. In Figure 5, we perform ablation study for diffusion Q-action gradient on synthetic data.
>
> ---
>
> Finally, we have outlined the list of updates made in the revised paper based on feedback from all the reviewers in [Overall Response](https://openreview.net/forum?id=FvQsk3la17&noteId=xFgYGltWDv) above. There, we provided a summary of new experiments and analysis that have been added to the revised paper.
>
> We hope we have addressed all of your questions/concerns. If you have any further questions, we would be more than happy to answer them and if you don’t, would you kindly consider increasing your score?
>
> [1] Ishfaq, Haque, et al. "Provable and Practical: Efficient Exploration in Reinforcement Learning via Langevin Monte Carlo." The Twelfth International Conference on Learning Representations. (2024)
>
> [2] Duan, Jingliang, et al. "DSAC-T: Distributional soft actor-critic with three refinements." arXiv preprint arXiv:2310.05858 (2023).
>
> [3] Lu, Cong, et al. "Synthetic experience replay." Advances in Neural Information Processing Systems 36 (2024).

---

> > ### Author Response · Authors · 2024-11-23
> > **Follow up**
> >
> > Dear Reviewer ETnf,
> >
> > Thank you so much for taking the time to review our paper. We sincerely appreciate your detailed feedback and appraisal of our work, and have carefully addressed your comments in our response and the revised manuscript. With only 3 days remaining for the discussion period, we would be grateful if you could acknowledge receipt of our responses and let us know if you have further questions. We are eager to engage in any further discussions if needed.
> >
> > If our responses have addressed all of your questions/concerns, would you kindly consider increasing your score?
> >
> > Thanks.\
> > Authors

---

> ### Comment · Reviewer_ETnf · 2024-11-27
>
> Thank you for your comments and addressing my concerns, however some main questions are not fully resolved. (For example, humanoid performance) I think 6 is a reasonable score at this stage so I will maintain the current score.

---

> > ### Author Response · Authors · 2024-11-27
> > **Thank you for your reply**
> >
> > We thank the reviewer for their comment. We would like to highlight that during the discussion phase so far, based on the suggestions from all the reviewers, we added lots of additional results and analysis as outlined in [Overall Response](https://openreview.net/forum?id=FvQsk3la17&noteId=xFgYGltWDv).
> >
> > We are more than willing to address any further concern the reviewer might have. If the reviewer could kindly specify the specific concerns they might have (along with the humanoid performance)  and what additional result or explanation from us might convince them for a higher score, we will try our best to address them.

---

> > > ### Author Response · Authors · 2024-11-30
> > > **Explanation on humanoid performance using Welch's t-test**
> > >
> > > We thank the reviewer again for their review.
> > >
> > > ---
> > >
> > > ### Welch’s t-test for LSAC and DSAC-T performance in humanoid
> > >
> > > To further address the reviewer’s question regarding the humanoid performance, we performed a Welch’s t-test experiment ([wikipedia link](https://en.wikipedia.org/wiki/Welch%27s_t-test)). Welch’s t-test is used to test the (null) hypothesis that two populations have equal means. Below we provide the terminal output of our Welch’s t-test on performance of LSAC and DSAC-T in mujoco humanoid environment:
> > >
> > > ```
> > > >>> import numpy as np, scipy.stats as st
> > > >>> lsac_humanoid
> > > [8631.37212366, 7509.77655114, 8898.04605687, 10444.20709681, 7340.40267768, 7340.43167725, 10543.44598872, 9109.56000168, 6924.74289811, 8712.35211212]
> > > >>> np.mean(lsac_humanoid)
> > > 8545.433718404
> > > >>> dsac_humanoid
> > > [9619.99523707, 9615.81430492, 10895.54379508, 6998.1882951, 7338.80670692, 9441.20751462, 8626.48317177, 11026.25789248, 8816.00723204, 7904.1112582]
> > > >>> np.mean(dsac_humanoid)
> > > 9028.241540820001
> > > >>> st.ttest_ind(lsac_humanoid, dsac_humanoid, equal_var = False)
> > > TtestResult(statistic=-0.8164981242172489, pvalue=0.424937425026642, df=17.91930902190612)
> > >
> > > ```
> > > Note that, in the Welch’s t-test, the p-value is $0.42$ which is **much higher** than $0.05$ and thus we cannot reject the null hypothesis and conclude that there is statistically no significant difference between the mean of the LSAC performance and the DSAC performance in mujoco humanoid environment.
> > >
> > >
> > > ---
> > >
> > > ### List of improvements since original submission
> > >
> > > We would like to highlight many additional experiments and analysis that we added during the discussion phase based on all reviewer’s suggestions.
> > >
> > > 1. We added two additional baselines in our Mujoco experiments (i) ensemble based REDQ and (ii) diffusion policy based QSM algorithm
> > > 2. Compared to 7 different DMC tasks with each run for 1 million steps in the original submission, we now have experiments for 12 DMC tasks each run for 3 million steps `(Fig 8, Table 2)`.
> > > 3. Ablation on distributional critic component in LSAC `(Fig 3 and Fig 16)`.
> > > 4. Comparison of normalized Q bias plot for ablation study of distributional critic in LSAC `(Fig 4 and Fig 17)`.
> > > 5. Exploration capability comparison of LSAC with baselines in maze environments through exploration density map and state coverage plots. `(Section 4.2, Fig 7, Fig 9 and Fig 10)`
> > > 6. Sensitivity analysis on the performance and training time of LSAC for different choices of
> > > update frequencies of diffusion model `(Fig 13)`.
> > > 7.  Distribution heatmaps of sampled actions $a$ in the online replay buffer $\mathcal D$, synthesized action samples $a_M$ in the diffusion buffer $\mathcal D'$, as well as $Q$ gradient-optimized actions $a_M \sim \nabla_a Q$ `(Fig 14)`.
> > > 8. Computational efficiency analysis `(Appendix F, Table 5, Fig 15)`.
> > > 9. Comparison of number of learnable parameters of LSAC and baselines `(Table 6)`.
> > >
> > > ---
> > >
> > > If the reviewer has any further concerns and could kindly specify them, we will try our best to address them. Finally, we are grateful for the reviewer’s continued positive support on the paper. Given our extensive additional experiment results as outlined above, and statistically thorough analysis for the humanoid performance of LSAC and DSAC-T, we would be immensely grateful, if the reviewer considers the work as worthy of a higher score.

---

> > > > ### Author Response · Authors · 2024-12-02
> > > > **Any last minute question before the discussion period ends**
> > > >
> > > > Dear Reviewer ETnf,
> > > >
> > > > As the discussion period is set to end soon, we kindly ask if you have any further questions or comments, or if there are any aspects of the paper that require additional clarification. We are grateful for your positive support and we want to ensure we address any remaining concerns.
> > > >
> > > > Thanks,\
> > > > Authors

---

### Official Review · Reviewer_TJsH · 2024-11-07

**Soundness:** 2
**Presentation:** 1
**Contribution:** 2
**Rating:** 3
**Confidence:** 3

**Summary:**

This paper aims to address inefficient exploration and bad sample usage in actor-critic algorithms for continuous control problems. They propose Langevin Soft Actor-Critic (LSAC) which uses adaptive Langevin Monte Carlo (aSGLD) to sample from a Q-value distribution, improving exploration by capturing uncertainty. A diffusion model is also applied to generate diverse synthetic state-action pairs, refined by Q-action gradients to focus on high-reward areas, thereby enhancing sample efficiency. The algorithm demonstrates improved or comparable empirical performance on MuJoCo and DeepMind Control Suite benchmarks compared to standard RL algorithms. An ablation study is also provided.

**Strengths:**

1. They propose a new method that models each Q(s, a) value as a distribution. Then, an LMC-based sampling algorithm is used to sample the posterior of the Q value, which encourages exploration. The accordingly distributional Bellman operator is used.
2. They propose using a diffusion model to generate more state-action pairs to update the critic function, where each pair's action is replaced with its Q action gradient updated ones. The use of generative models helps increase the sample efficiency.
3. Thorough experiments are conducted, with the algorithm implemented on two benchmarks, demonstrating improved or comparable empirical performance.

**Weaknesses:**

My primary concerns are with novelty and clarity.

**Contribution:**
Based on my reading of this submission, this work extends Langevin Monte Carlo (LMC) [2] from discrete to continuous action spaces (e.g., MuJoCo tasks). It builds on [2] with the same critic modeling as [1] and integrates a diffusion model for generating synthetic replay data [4].

The main novelties introduced in this work seem to include:
1. extending the use of multiple Q-values from Langevin-gradient-based Bayesian neural learning [3] to RL.
2. using gradient ascent-refined actions in diffusion-based policy optimization [5] to improve data diversity in experience replay [4].
3. reducing the number of aSGLD updates for the critic in [2] from $O(K)$ to a single iteration. However, I am uncertain whether this improvement arises from the diffusion model generating more data, which increases the size of batch in Algorithm 2.

Despite the outlined contributions, I have several **questions** regarding the arguments and claims made in the submission:
1. the paper claims multiple critics act as parallel tempering to address slow mixing in LMC. However, LMC is described as an optimization rule, not a distribution approximation. Does the issue exist in the LMC-inspired critic update? And does the gain in Figure 6 come from parallel tempering or simply reduced TD target variance, as in SAC, since all critics share the same temperature?
2. from the ablation study, the distributional critic modeling seems most impactful. Is this due to exploration from the variance in the Gaussian Q-values distribution? My previous question regarding the distributional TD loss is that: for the distributional TD loss $L(\psi)$ and normal deterministic TD loss $TD_c(\psi)$, $L(\psi) = \frac{TD_c(\psi)}{\sigma_{\psi}} + \log(\sigma_{\psi})$ as shown in Eq(15), higher $\sigma_{\psi}$ increases exploration (second term) but reduces TD loss (first term). Therefore, I am curious whether this $\sigma_{\psi}$-$TD_c(\psi)$ trade-off inside the training of $L(\psi) $ is related to the improvement in Figure 3,4.


**Presentation:**
 many techniques or claims are applied without sufficient interpretation or explanation. For example, in section 3, the paragraph **Distributional Critic Learning with Adaptive Langevin Monte Carlo** mostly explains [1] and [2], and Equation (9) is the same as lines 9-11 of Algorithm 2 in [2]. It is better to move them to the preliminary section. As a reader, it is hard to understand the paper without reading other references. When presented in the Algorithm Design section, I would expect the work to provide a self-contained explanation of its logic and methodology.

I truly appreciate that the author included comparisons of running time and parameter numbers. But when a paper adapts old algorithms to a new task, I would expect this work can provide deeper analysis and insights specific to the task at hand. This would enhance the impact.


[1] Jingliang Duan, Wenxuan Wang, Liming Xiao, Jiaxin Gao, and Shengbo Eben Li. DSAC-T: Distributional soft actor-critic with three refinements.

[2] Haque Ishfaq, Qingfeng Lan, Pan Xu, A Rupam Mahmood, Doina Precup, Anima Anandkumar, and Kamyar Azizzadenesheli. Provable and practical: Efficient exploration in reinforcement learning via Langevin Monte Carlo. In The Twelfth International Conference on Learning Representations

[3] Rohitash Chandra, Konark Jain, Ratneel V Deo, and Sally Cripps. Langevin-gradient parallel tempering for bayesian neural learning

[4] Cong Lu, Philip Ball, Yee Whye Teh, and Jack Parker-Holder. Synthetic experience replay. Advances in Neural Information Processing Systems, 36, 2024.

[5] Long Yang, Zhixiong Huang, Fenghao Lei, Yucun Zhong, Yiming Yang, Cong Fang, Shiting Wen, Binbin Zhou, and Zhouchen Lin. Policy representation via diffusion probability model for reinforcement learning.

**Questions:**

see weakness

---

> ### Author Response · Authors · 2024-11-20
>
> We thank the reviewer for their valuable time and effort in providing detailed feedback on our work. We hope our response will fully address all of the reviewer's points.
>
> ---
>
>
> ### Adaptive preconditioner $\zeta_k$ in Equation (8) and (9).
>
> The adaptive preconditioner $\zeta_k$ is used to accelerate the convergence of Langevin Monte Carlo (LMC) in the presence of pathological curvatures and saddle points. We explained the role of adaptive preconditioner in Line 233-238:
>
> ```
> In place of vanilla LMC described in Equation 2, following Ishfaq et al. (2024a); Kim et al. (2022), we use adaptive Stochastic Gradient Langevin Dynamics (aSGLD), where an adaptively adjusted bias term is included in the drift function to enhance escape from saddle points and accelerate the convergence to the true Q posterior, even in the presence of pathological curvatures and saddle points which are common in deep neural network (Dauphin et al., 2014).
>
> ```
>
> The adaptive preconditioner is inspired from the Adam optimizer [1] and thus modeled as Equation (9). The theoretical properties of such Adam inspired adaptive preconditioner when used in supervised learning is analyzed in [2].
>
> ---
>
> ### Entropy coefficient $\alpha$ in Equation (12).
>
> The entropy coefficient $\alpha$ update rule in Equation (12) is a standard component of SAC implementation [3] for automating the entropy adjustment in maximum entropy RL. The rationale behind this update rule is discussed in detail in Section 5 and Section 6 in [3]. We refer the reviewer to Equation (17), Equation (18) and the step “Adjust temperature” in Algorithm 1 (Soft Actor-Critic) of [3].
>
>
> ---
>
>
> ### Complexity of the algorithm and the pipeline.
>
> We would like to highlight that in essence, LSAC has 3 main components: (1) LMC based distributional Q sampler, (2) Synthetic data generation using a diffusion model which is done once every `1e4` steps. (3) Parallel tempering whose implementation is very similar to standard ensemble based approaches. Once these components are considered (and implemented) individually, after piecing them together, we hope it would not look as complex as it might seem at first glance.
>
> We would also like to mention that using parallel LMC as opposed to vanilla LMC should be seen as a strength which are discussed in detail from Line 267 - 280.
>
>
> ---
>
>
> ### We added computational efficiency analysis and comparison of network capacity.
>
> Thanks for your suggestion to add time efficiency analysis. We added a comprehensive time/computational efficiency analysis in Appendix F. The comprehensive plot can be found in Figure 15 and a table in Table 5. We also added a detailed table comparing the number of learnable parameters for different algorithms in Table 6.
>
> ---
>
>
> ### Detailed analysis using ablation studies and insights.
>
> In Section 4.1, we showed several ablation studies of key components of LSAC in the original submission and added additional ablation studies in the revised version. On Figure 3 and Figure 16, we show the impact of using distributional critics in LSAC. We compare LSAC with distributional critics and a version of LSAC where the distributional critics are replaced by  standard critics implementation (such as in SAC). Furthermore, in Figure 4 and Figure 17, we show how distributional critic in LSAC helps mitigating overestimation bias. In Figure 6 (a), we show the impact of parallel tempering with different numbers of parallel critics. Note that, in this Figure the case with $|\Psi| = 1$ is equivalent to not having any parallel tempering. In Figure 6 (b), we compare the performance when critic is updated using LMC based aSGLD sampler for Q values vs when the critic is updated using the standard Adam optimizer. In Figure 5, we perform ablation study for Q-action gradient on synthetic data. Figure 2 shows sensitivity analysis of different hyperparameters.  Each of these experiments are discussed in detail in Section 4.1.
>
> Finally in the revised version, in Section 4.2 we add an additional experiment that compares the exploration capability of LSAC with other baselines using exploration density maps in maze environments.
>
> We would highly appreciate it if the reviewer could kindly let us know their suggestion on how our current insights and analysis can be further developed. We will try our best to incorporate those suggestions.

---

> > ### Author Response · Authors · 2024-11-20
> > **Continuation of response (part 2)**
> >
> > ---
> >
> >
> > ### Answers to questions.
> >
> >
> > * > Is $Q_{\psi}(s,a)$ deterministic or stochastic? How is $Q_{\psi}(s,a)$ sampled in Equation (11)? Given that $Z_{\psi}(\cdot| s, a) = N(Q_{\psi}(s, a), \sigma_{\psi}(s, a))$ is defined as the distribution of the $Q(s,a)$ value, I would expect the $Q_{\psi} + \text{noise} \cdot \sqrt{\sigma}$ term to represent the Q-value.
> >
> > Following max-entropy RL, we aim to update policy using the objective
> > $$\pi_{\text{new}} = \underset{\pi}{\operatorname{argmax}} E_{s \sim \rho_\pi, a \sim \pi } [ Q^{\pi_{\text{old}}}(s,a) - \alpha \log \pi (a |s)]$$ the parameterized version of which is given by Equation (11). Note that, here $Q_\psi$ is parameterized version of soft Q-value. As defined in Line 119-128, for any policy $\pi$, we have soft Q value $Q^\pi(s,a)$ and random variable soft state action return $Z^\pi(s,a)$ with $Q^\pi(s,a) = E[Z^\pi(s,a)]$. Thus, in our notation and definition $Q_\psi$ which parameterizes soft Q value $Q^\pi$ is not a random variable but rather the mean of the distribution over soft state action return $Z$. Thus for a fixed state-action pair $(s,a)$, $Q_\psi$ is deterministic.
> >
> > * **Question on role of LMC** : The role of LMC is to approximately sample from this exponential form distribution. Here, direct sampling is not possible as partition function $Z$ is unknown. The role of LMC is to perform approximate TS by sampling $Q_\psi$ from this exponential form distribution. In line 9 (line 12 in original submission), we are sampling $\psi^{(i)}$ which is parameter (which itself is updated using LMC in Line 8 of Algorithm 1) of one of the critics from the list of parallel critics. Essentially, in line 9, we are uniformly sampling a “critic” from the set of parallel critics. So, these are two different things.
> >
> > * **Question on TD loss** : The KL loss in Equation (6) can be thought of as a distributional variant of standard TD loss. Equation (15) is a refined form of the KL loss resulting from Proposition B.1. To our understanding, this does not imply any trade-off between the standard TD loss and the KL loss used in our case.
> >
> > ---
> >
> > Finally, we have outlined the list of updates made in the revised paper based on feedback from all the reviewers in [Overall Response](https://openreview.net/forum?id=FvQsk3la17&noteId=xFgYGltWDv) above. There, we provided a summary of new experiments and analysis that have been added to the revised paper.
> >
> >
> > We hope we have addressed all of your questions/concerns. If you have any further questions, we would be more than happy to answer them and if you don’t, would you kindly consider increasing your score?
> >
> > [1] Kingma, Diederik P. "Adam: A method for stochastic optimization." arXiv preprint arXiv:1412.6980 (2014).
> >
> > [2] Kim, Sehwan, Qifan Song, and Faming Liang. "Stochastic gradient Langevin dynamics with adaptive drifts." Journal of statistical computation and simulation 92.2 (2022): 318-336.
> >
> > [3] Haarnoja, Tuomas, et al. "Soft actor-critic algorithms and applications." arXiv preprint arXiv:1812.05905 (2018).

---

> ### Author Response · Authors · 2024-11-23
> **Follow up**
>
> Dear Reviewer TJsH,
>
> Thank you so much for taking the time to review our paper. We sincerely appreciate your detailed feedback and appraisal of our work, and have carefully addressed your comments in our response and the revised manuscript. With only 3 days remaining for the discussion period, we would be grateful if you could acknowledge receipt of our responses and let us know if you have further questions. We are eager to engage in any further discussions if needed.
>
> If our responses have addressed all of your questions/concerns, would you kindly consider increasing your score?
>
> Thanks.
>
> Authors

---

> > ### Comment · Reviewer_TJsH · 2024-11-24
> >
> > I reviewed the revision. While some of my concerns have been addressed, most remain unresolved. Please see the comment. I will maintain the current score. I appreciate the authors for providing the comparison of runtime and parameter counts, so I slightly lower my confidence level.

---

> > > ### Author Response · Authors · 2024-11-26
> > > **Response to updated review (Part 1)**
> > >
> > > We thank the reviewer for their response. We noticed that the `Weaknesses` section as well as the `Questions` sections of the review have largely been rewritten with many earlier questions being removed after our initial response to the original review. Before addressing points raised in the *updated* review, we want to emphasize that, in our earlier response, we addressed every point raised in the weakness and questions section in the original review. We hope our earlier answers addressed the concerns for the points and questions that have been removed by the reviewer from the original review.
> > >
> > > ---
> > >
> > >
> > > ### Contribution
> > >
> > > >  *Based on my reading of this submission, this work extends Langevin Monte Carlo (LMC) [2] from discrete to continuous action spaces (e.g., MuJoCo tasks).*
> > >
> > > We believe considering this work just as an extension of LMC from discrete to continuous action spaces is not a fair assessment. As the other reviewers have acknowledged, this is the first successful application of LMC based Thompson sampling in RL with continuous control setting and continuous action space. Moreover, just because LMC based Thompson sampling was proposed for discrete action space, does not mean extending it to continuous action space is trivial. Generally speaking, extending applicability of any algorithm specifically designed for discrete action space to continuous action spaces poses several challenges, many of which were outlined in our introduction section  as well.
> > >
> > > ---
> > >
> > > ### Novelty
> > >
> > > While the core idea of Q update in LSAC is to perform Langevin Monte Carlo (LMC) update [2]  with distributional critic [1], the idea of combining them and the resulting benefit is non-trivial and non-obvious. Moreover, LSAC has several other novel design choices that make it significantly different from [5] and DSAC-T [1]. For example, instead of standard LMC update from [2], we use parallel tempering with LMC for exploring different modes of the Q-posterior. As the other reviewers (Reviewer ETnF, Reviewer iNev and Reviewer j9KN) mentioned, this work is the first successful application of LMC based Thompson sampling in continuous action space. Moreover as the Reviewer ETnf mentioned, “*methodological choices in LSAC are grounded in theoretical analysis presented in the paper.*” Moreover, as the Reviewer j9KN mentioned, “*compared with diffusion as a policy, this paper does not have the risk of too long sampling time in actual use.*”
> > > Thus, we believe that the contribution of this work to the RL literature would be significant.
> > >
> > >
> > > ---
> > >
> > >
> > > ### Number of aSGLD update vs diffusion model data
> > >
> > > > *reducing the number of aSGLD updates for the critic in [2] from O(K)  to a single iteration. However, I am uncertain whether this improvement arises from the diffusion model generating more data, which increases the size of batch in Algorithm 2.*
> > >
> > > We found this statement to be ambiguous. Could you please clarify? We don’t see any connection between the number of aSGLD updates for the critic and the improvement contributed by diffusion model generating synthetic data. They both serve two different purposes. Reducing the number of aSGLD updates from $O(K)$ to a single iteration is a design choice to speed up training. On the contrary the purpose of the diffusion model generated synthetic data is to enhance diversity in the state-action pairs and increase the UTD ratio for critic updates as mentioned in Line 282-284. Also, we do not increase the size of the batch as the reviewer thought. In fact, as we explained in Appendix E.3 (Line 1144-1146), we construct a batch size of 256 using an equal amount of data points (128) sampled from both true replay buffer and the synthetic replay buffer. As shown in Table 4, our effective batch size (256) is the same as what other baselines use.
> > >
> > >
> > > ---
> > >
> > > ### Slow mixing in LMC
> > >
> > > > *the paper claims multiple critics act as parallel tempering to address slow mixing in LMC. However, LMC is described as an optimization rule, not a distribution approximation. Does the issue exist in the LMC-inspired critic update?*
> > >
> > > As described in Line 135 -143, LMC is a means to sample from a distribution through an optimization style update rule. In Line 231-232, we state that:
> > >
> > > “In Appendix B we show that, under some mild assumptions, the posterior over $Q_\psi$ is of the form $\exp (-L(\psi)) / Z$, where $Z$ is the partition function”
> > >
> > > The goal of the LMC based update rule is to sample from this distribution. Moreover, as we explained in Line 89-97, slow mixing is a common downside of naive LMC based methods. In fact, slow mixing is a well-known problem in LMC when the target distribution has certain characteristics such as high dimensional parameter space, multimodal distribution,  ill-conditioned energy landscape which are common when using deep neural networks. All of these characteristics are present in the LMC inspired critic update.
> > >
> > >
> > > ---

---

> > > > ### Author Response · Authors · 2024-11-26
> > > > **Response to updated review (Part 2)**
> > > >
> > > > ---
> > > >
> > > >
> > > > ### Gain in Figure 6
> > > >
> > > > > *does the gain in Figure 6 come from parallel tempering or simply reduced TD target variance, as in SAC, since all critics share the same temperature?*
> > > >
> > > > Figure 6 (a) shows the effect of parallel tempering for different numbers of samplers (i.e. replicas). Note that when $|\Psi_Q| = 1$, it's the same as not using parallel tempering. Thus, it clearly shows the benefit of using parallel tempering. On the other hand,
> > > > Figure 6(b) shows the difference when  distributional critics are optimized using aSGLD (which performs Thompson sampling over Q posterior) vs when distributional critics are simply optimized using the standard Adam optimizer. Since we use multiple critics, using Adam on multiple distributional critics can be thought of as something similar to a distributional variant of bootstrapped DQN [6]. So, by experiment design, it should be evident that the performance gain in Figure 6(b) comes from LMC based Thompson sampling.
> > > >
> > > >
> > > > ---
> > > >
> > > > ### Ablation study of distributional critic modeling
> > > >
> > > > > *from the ablation study, the distributional critic modeling seems most impactful. Is this due to exploration from the variance in the Gaussian Q-values distribution?*
> > > >
> > > > The usage of distributional critic while using LMC based update for critic stabilizes learning by mitigating overestimation bias. The impact of using LMC with distributional critics in reducing overestimation bias has been shown in ablation study plots in Figure 4 and Figure 17. The gain is attributed to reducing overestimation bias.
> > > >
> > > > As we explained in our original response, we are not using Gaussian Q value distribution nor the variance to drive exploration. In fact, we model and parameterize the value distribution function $\mathcal Z_\psi$ using Gaussian. Note that here, value distribution function $\mathcal {Z}_\psi$ defined in Line 199-200, models the distribution of the random variable $Z^\pi(s,a)$ where we refer to $Z^\pi(s,a)$ as soft state action return (defined in Line 196).
> > > >
> > > >
> > > >
> > > > ---
> > > >
> > > > ### Question about distributional TD loss
> > > >
> > > > The reviewer defined $TD_c(\psi)$  as standard deterministic TD loss. But in Eq (15), we are not using a deterministic TD loss $TD_c(\psi)$ as defined by the reviewer. Note that in Eq (15), we have (without considering the expectation over replay buffer data), $L_{\mathcal Z}(\psi) = \frac{(T^{\pi}Z - Q_\psi)^2}{\sigma_{\psi}^2} + \log(\sigma_{\psi})$. Here $Z$ is soft state-action return and a random variable as defined in Line 196, and as defined in Eq (5), $T^{\pi}Z$ is also a random variable. Thus the term $(T^{\pi}Z - Q_\psi)^2$ in $L_{\mathcal Z}(\psi)$ is not deterministic as opposed to what the reviewer is saying. Thus the equation the reviewer wrote for $L(\psi)$  in their review is incorrect and not equivalent to what our Eq (15) says.
> > > >
> > > > Finally, we are not using $\sigma_\psi$ to drive up exploration. Eq (15), is a simplified expression for the loss function in Eq (13) and Eq (14). As shown in Eq (18), the distribution over Q posterior can be shown as $ p( \psi | B) \propto \exp( - L_{\mathcal Z}(\psi))$. The goal of LMC based Thompson sampling is to sample $\psi$ from this distribution which in effect samples a Q function. This sampling process of Q function  is what drives the exploration in Thompson sampling.
> > > >
> > > > ---
> > > >
> > > > ### Presentation
> > > >
> > > > > *many techniques or claims are applied without sufficient interpretation or explanation.*
> > > >
> > > > Would you kindly be more specific on which parts felt to have insufficient interpretation and explanation? We will do our best to address them.
> > > >
> > > > > *in section 3, the paragraph Distributional Critic Learning with Adaptive Langevin Monte Carlo mostly explains [1] and [2], and Equation (9) is the same as lines 9-11 of Algorithm 2 in [2]. It is better to move them to the preliminary section.*
> > > >
> > > > The paragraph Distributional Critic Learning with Adaptive Langevin Monte Carlo does not just explain [1] and [2]. In fact, it explains how posterior sampling (i.e. Thompson sampling)  over distributional critic can be performed using adaptive LMC. The purpose of the paragraph is to show the synergy between them. Also, Eq (8) and (9) are adaptive LMC update rules which first appeared in the context of supervised learning in [7]. Even though adaptive LMC was also used in [2] in a DQN-like algorithm, we are using it in an actor-critic based algorithm. Most importantly, LMC is a standard procedure and thus it seems unfair to characterize it as ”*same as lines 9-11 of Algorithm 2 in [2]*”. Moreover, given we have a dedicated paragraph on standard Langevin Monte Carlo in the preliminary section, we feel it makes more sense to keep Eq (8) and (9) under the paragraph Distributional Critic Learning with Adaptive Langevin Monte Carlo. Also, we feel it helps the Algorithm design section to be more self-contained which the reviewer has also emphasized.
> > > >
> > > >
> > > > ---

---

> > > > > ### Author Response · Authors · 2024-11-26
> > > > > **Response to updated review (Part 3)**
> > > > >
> > > > > ---
> > > > >
> > > > > ### On deeper analysis and insights
> > > > >
> > > > > > *I would expect this work can provide deeper analysis and insights specific to the task at hand. This would enhance the impact.*
> > > > >
> > > > > Along with proposing the first LMC based Thompson sampling algorithm that can work in continuous action spaces, we also provided extensive analysis and insights for LSAC. Here is a list of the analysis provided:
> > > > >
> > > > > 1. Experiment in 6 Mujoco tasks with 8 baselines `(Fig 1, Table 1)`.
> > > > > 2. Experiment in 12 DMC tasks with 8 baselines `(Fig 8, Table 2)`.
> > > > > 3. Sensitivity analysis for the bias factor $a$, learning rate $\eta_Q$ and inverse temperature. $\beta_Q$ in adaptive LMC update rule `(Fig 2)`.
> > > > > 4. Ablation on distributional critic component in LSAC `(Fig 3 and Fig 16)`.
> > > > > 5. Comparison of normalized Q bias plot for ablation study of distributional critic in LSAC `(Fig 4 and Fig 17)`.
> > > > > 6. Ablation on synthetic experience replay with action gradient ascent `(Fig 5)`
> > > > > 7. Ablation on parallel critics `(Fig 6a)`
> > > > > 8. Ablation on LMC based critic learning vs standard critic learning through Adam optimizer `(Fig 6b)`
> > > > > 9. Learning stability analysis `(Fig 18)`
> > > > > 10. Exploration capability comparison of LSAC with baselines in maze environments through exploration density map and state coverage plots. `(Section 4.2, Fig 7, Fig 9 and Fig 10)`.
> > > > > 11. Theoretical insight `(Appendix B)`.
> > > > > 12. Sensitivity analysis on the performance and training time of LSAC for different choices of
> > > > > update frequencies of diffusion model `(Fig 13)`.
> > > > > 13. Distribution heatmaps of sampled actions $a$ in the online replay buffer $\mathcal D$, synthesized action samples $a_M$ in the diffusion buffer $\mathcal D'$, as well as $Q$ gradient-optimized actions $a_M \sim \nabla_a Q$ `(Fig 14)`.
> > > > > 14. Computational efficiency analysis `(Appendix F, Table 5, Fig 15)`.
> > > > > 15. Comparison of number of learnable parameters of LSAC and baselines `(Table 6)`.
> > > > >
> > > > > We would appreciate it if the reviewer has any further suggestion for deeper analysis of LSAC and we would try our best to incorporate those suggestions.
> > > > >
> > > > >
> > > > >
> > > > > ---
> > > > > We hope we have addressed all of your questions/concerns. If you have any further questions, we would be more than happy to answer them and if you don’t, would you kindly consider increasing your score?
> > > > >
> > > > >
> > > > > [1] Jingliang Duan, Wenxuan Wang, Liming Xiao, Jiaxin Gao, and Shengbo Eben Li. DSAC-T: Distributional soft actor-critic with three refinements.\
> > > > >
> > > > > [2] Haque Ishfaq, Qingfeng Lan, Pan Xu, A Rupam Mahmood, Doina Precup, Anima Anandkumar, and Kamyar Azizzadenesheli. Provable and practical: Efficient exploration in reinforcement learning via Langevin Monte Carlo. In The Twelfth International Conference on Learning Representations\
> > > > >
> > > > > [3] Rohitash Chandra, Konark Jain, Ratneel V Deo, and Sally Cripps. Langevin-gradient parallel tempering for bayesian neural learning\
> > > > >
> > > > > [4] Cong Lu, Philip Ball, Yee Whye Teh, and Jack Parker-Holder. Synthetic experience replay. Advances in Neural Information Processing Systems, 36, 2024.\
> > > > >
> > > > > [5] Long Yang, Zhixiong Huang, Fenghao Lei, Yucun Zhong, Yiming Yang, Cong Fang, Shiting Wen, Binbin Zhou, and Zhouchen Lin. Policy representation via diffusion probability model for reinforcement learning.\
> > > > >
> > > > > [6] Osband, Ian, et al. "Deep exploration via bootstrapped DQN." Advances in neural information processing systems 29 (2016).\
> > > > >
> > > > > [7] Kim, Sehwan, Qifan Song, and Faming Liang. "Stochastic gradient Langevin dynamics with adaptive drifts." Journal of statistical computation and simulation 92.2 (2022): 318-336.

---

> ### Author Response · Authors · 2024-11-27
> **Looking forward to your feedback**
>
> Dear Reviewer TJsH,
>
> We greatly appreciate your time and effort, as we understand you may have a busy schedule. As the paper revision deadline is approaching, we hope our updates and answers above address all your concerns, and if they do, we would be grateful if you could consider increasing your score. Your support means a lot to us.
>
> If you still have any remaining concerns or questions, could you please specify them? We will try our best to address them as quickly as possible.
>
> Thanks in advance,\
> Authors

---

> > ### Author Response · Authors · 2024-12-01
> > **Follow up**
> >
> > Dear Reviewer TJsH,
> >
> > We greatly appreciate your time and effort to review our paper. As we are approaching the end of the discussion period with only 2 days remaining, we hope our responses have addressed all your concerns. If so, could you please acknowledge our response and consider increasing your score? We would really appreciate your support.
> >
> > If you still have any remaining concerns, could you please specify them? As we consistently mentioned in our earlier messages, we will try our best to address them.
> >
> > Thanks,\
> > Authors

---

### Author Response · Authors · 2024-11-20
**Overall Response**

We would like to thank all the reviewers for their insightful and detailed reviews and comments. We are grateful that the reviewers have mostly recognized our major contributions and provided insightful suggestions to improve the paper. We have addressed the comments from the reviewers and revised the manuscript accordingly with several new experiments and analysis. We summarize the main updates done in the revised paper here:

---

### 1. Adding additional baselines in the MuJoCo experiment.
We added two new baselines in our MuJoCo experiment. (i) ensemble based REDQ algorithm [1] and (ii) diffusion policy based QSM algorithm [2].

---

### 2. Experiments in additional DMC tasks with a higher number of steps.
In our original submission, we had experiments on 7 different DMC tasks and each task was trained for 1 million steps only. In the revised paper, we added experiments on 5 more difficult DMC tasks and each task is now trained for 3 million steps. We report the result in Appendix C, Table 2 and Figure 8. LSAC outperforms other baselines in 9 out of 12 tasks.

---

### 3. Ablation study on distributional critic.

To understand the impact of distributional critic in LSAC performance, we run ablation studies where we replace our distributional critic with standard critic implementation in SAC. We observe the performance impact in Figure 3 and Figure 16. We also report how distributional critic in LSAC can reduce overestimation bias in Figure 4 and Figure 17.


---

### 4. Understanding exploration ability of LSAC using exploration density map.
In Section 4.2, we study the exploration ability of LSAC in comparison to other baselines in two types of maze environments. We plot exploration density map and state coverage of LSAC and baseline algorithms (Figure 7, Figure 9 and Figure 10). We find that LSAC has superior exploration ability.

---

### 5. Related Work
We moved the Related Work section to  Appendix A  due to space constraint in the main paper. We added a new paragraph in related work where we discuss recent progress in online RL with diffusion based policy learning.

---

### 6. Sensitivity analysis on update frequencies for diffusion model.

We ran a sensitivity analysis on the performance of LSAC for different update frequencies for the diffusion model. We present the result in Figure 6 (c ) and Figure 13. We observe that increasing the update frequency doesn’t really impact on the average return much while increasing the training time significantly.

---

### 7. Analysis of sample distribution for diffusion generated data using heatmap.
In Figure 14, we show distribution heatmaps of sampled actions from online replay buffer, diffusion synthesized action samples, and action gradient refined synthetic actions. From the figure, we see that when we perform action gradient on synthetic diffusion generated action, the resulting distribution of refined action matches closely with the action distribution of real online replay buffer.


---

### 8. We added time efficiency analysis and comparison of network capacity.
 We added a comprehensive time efficiency analysis in Appendix F. The comprehensive plot can be found in Figure 15 and a table in Table 5. We also added a detailed table comparing the number of learnable parameters for different algorithms in Table 6.

---


We look forward to hearing back from the reviewers and we will be happy to answer any further questions the reviewers might have.



[1] Chen, Xinyue, et al. "Randomized Ensembled Double Q-Learning: Learning Fast Without a Model." International Conference on Learning Representations. (2021)


[2] Psenka, Michael, et al. "Learning a Diffusion Model Policy from Rewards via Q-Score Matching." Forty-first International Conference on Machine Learning. (2024)

---

### Meta-Review · Area_Chair_r9kv · 2024-12-19

**Metareview:**

Langevin Soft Actor-Critic: Efficient Exploration through Uncertainty-Driven Critic Learning

Summary: The paper introduces Langevin Soft Actor-Critic (LSAC), a reinforcement learning algorithm designed to improve exploration in continuous control tasks. LSAC addresses limitations in traditional actor-critic methods, such as poor sample efficiency and lack of principled exploration. The algorithm leverages Langevin Monte Carlo (LMC) for uncertainty-driven updates to the critic, combined with parallel tempering to explore multimodal posterior distributions of the Q function. Additionally, it introduces diffusion-based synthetic state-action samples, refined through Q action gradients, to enhance critic learning. Theoretical contributions include a distributional adaptive Langevin Monte Carlo framework and insights into reducing overestimation biases. Experimentally, LSAC demonstrates improved performance across MuJoCo and DeepMind Control Suite benchmarks compared to state-of-the-art baselines.

Comments: We received four expert reviews, with the scores  3, 6, 6, 8, and the average score is 5.75.

Reviewers are generally positive, with the exception of one review by Reviewer TJsH. I have carefully gone through this reviewer's questions and the authors’ detailed responses. Based on my reading, I believe that the authors have clearly addressed all the concerns and comments raised by this reviewer.

Other reviewers are positive about the contribution of the paper. They emphasized the novelty of methodological contributions which combine LMC, parallel tempering, and diffusion-synthesized action refinement to improve exploration in continuous control RL. Reviewers are also satisfied with empirical results that show LSAC outperforms or matches state-of-the-art baselines on a range of benchmarks, including MuJoCo and DeepMind Control Suite. Reviewers also appreciated the theoretical insights provided in the paper.

At the same time, the reviewers have provided multiple comments to strengthen the paper. For example, one reviewer pointed out that the parallel critics and diffusion buffers add to implementation complexity and computational cost and has asked for a detailed discussion about this. Reviewers have also asked for more detailed related works discussions, especially about works on combining diffusion models and RL. Reviewers have also asked for additional ablation studies.

Overall, the paper makes a meaningful contribution to the field of RL by introducing LSAC, a novel and theoretically grounded algorithm for efficient exploration in continuous control tasks. Despite some limitations in scalability and experimental depth, these weaknesses do not outweigh the overall strength and potential impact of the work.

**Additional Comments On Reviewer Discussion:**

Please see the "Comments" in the meta-review.

---

### Decision · Program_Chairs · 2025-01-22

Accept (Poster)